



# Catchment transit time sensitivity to the type of SAS function for unsaturated zone and groundwater

Hatice Türk[1], Christine Stumpp[1], Markus Hrachowitz[2], Peter Strauss[3], Günter Blöschl[4], and Michael Stockinger[1]

[1]BOKU University, Institute of Soil Physics and Rural Water Management, Department of Landscape, Water and Infrastructure, Muthgasse 18, 1190 Vienna, Austria
[2]Department of Water Management, Faculty of Civil Engineering and Geosciences, Delft University of Technology, Stevinweg 1, 2628 CN Delft, the Netherlands
[3]Institute for Land and Water Management Research, Federal Agency for Water Management, Petzenkirchen, Austria
[4]Vienna University of Technology, Institute of Hydraulic Engineering and Water Resources Management, Karlsplatz 13, 1040 Vienna, Austria

**Correspondence:** Hatice Türk (hatice.tuerk@boku.ac.at)

**Abstract.** Preferential flow paths in hydrological systems (e.g., macropores or subsurface pipe networks) facilitate rapid water and solute transport, leading to fast streamflow responses and markedly short transit times. While such preferential flow processes are well known in the unsaturated zone and groundwater, it remains uncertain whether catchment-scale isotope-based transport models can accurately represent these fast groundwater flow processes. In this study, we tested the hypothesis that

preferential discharge of young groundwater is significant and can be captured by selecting specific StorAge Selection (SAS) functions, i.e., functions that specify if young or old water leaves a storage, at the catchment scale. We systematically compared multiple SAS parameterisations for the unsaturated zone and groundwater using a catchment scale transport model and long-term measurements of hydrogen isotopes ($\delta^2$H) data from two headwater catchments (Hydrological Open Air Laboratory, HOAL, catchment in Austria and Wüstebach catchment in Germany). The results indicated that $\delta^2$H ratios in streamflow had

sufficient information content to identify preferential flow in the unsaturated zone. However, $\delta^2$H ratios in streamflow were insufficient to constrain or confirm preferential flow in groundwater, as any seasonal variation of $\delta^2$H in pore water was largely dampened by the catchments' substantial passive groundwater storage volumes. This was further confirmed as the observed attenuated $\delta^2$H signal in streamflow could only be simulated when the volume ratio between active and passive groundwater storage was $< 1\,\%$. This damping effect affected the estimation of the longer tails ($100 < T < 1000$ days) of the transit time dis-

tributions, making it challenging to estimate how much of the streamwater actually is older than 100 days. In addition, weekly resolution $\delta^2$H measurements led to deceptively high-performance metrics (e.g., Nash–Sutcliffe Efficiency), even when key model parameters for groundwater age selection —such as young- versus old-water selection preferences—remain poorly constrained. As a result, the variation in the estimation of the fraction of stream water younger than 1000 days was approximately 20 % in the HOAL and 23 % in the Wüstebach catchments due to the SAS function shape holding similar model performance.

These findings underscore the need for complementary data sources, such as multiple tracers, high-frequency sampling, or groundwater-level monitoring, to better constrain preferential flow processes and to reduce uncertainty in catchment-scale water transit time modelling.



## 1 Introduction

Groundwater plays a crucial role in the hydrological cycle and in sustaining streamflow in dry periods, thus regulating the tim-
ing and quality of water reaching streams (van der Velde et al., 2011; Hamilton, 2012; Kaandorp et al., 2018b). The movement
of precipitation through the soil matrix into the groundwater and eventually to the stream spans a wide range of timescales:
from rapid responses over days to months (Kaandorp et al., 2018a) to slower contributions over years to decades (Visser et al.,
2009; Stewart and Morgenstern, 2016; Wang et al., 2025). This variability in flow timescales is driven by many factors, in-
cluding catchment topology and subsurface flow path heterogeneity, which, in turn, leads to spatial and temporal variability in
stream water sources and chemical composition (McGuire and McDonnell, 2006; Hamilton, 2012; Kaandorp et al., 2018b). In
the light of these complexities, previous studies have long underscored that preferential flow pathways in both partially (Beven
and Germann, 1982; Weiler et al., 2003; Klaus et al., 2013) and fully saturated porous media (Bianchi et al., 2011) lead to
fast and localised water flow and solute transport, which have the potential to alter stream chemical composition dramatically.
Such preferential flow is widely acknowledged in groundwater hydrology (Berkowitz et al., 2006; Hansen and Berkowitz,
2020a; Berkowitz and Zehe, 2020; Hansen and Berkowitz, 2020b; Zehe et al., 2021), typically referred to as "non-Fickian" or
"anomalous" flow in the groundwater community (Berkowitz and Zehe, 2020; Hansen and Berkowitz, 2020a). While explic-
itly represented in many dedicated groundwater models (e.g. Berkowitz and Zehe 2020), it remains uncertain whether simpler,
top-down catchment-scale, isotope-based transport models can meaningfully detect and quantify preferential groundwater flow
pathways.
Water molecules entering at different locations of a catchment follow different flow paths and take different times to exit the
system again via streamflow or evaporation (transit time, TT). The statistical distribution of these transit times is referred
to as the transit time distribution (TTD). The transit time of water reflects the key information about how quickly water
moves through a catchment (Beven, 2006; Rinaldo et al., 2015; Benettin and Bertuzzo, 2018); hence, how quickly solutes
are transported through the surface, subsurface, and eventually to the stream. Despite their usefulness in studying water flow
through catchments, TTs cannot be measured directly and are generally inferred using hydrologic models and catchment-wide
input-output signals of tracers, such as water stable isotopes ($\delta^2$H, $\delta^{18}$O). For the quantification of flow processes and transit
times, many studies have integrated hydrometeorological data and applied tracer-based modelling (e.g., Birkel et al. 2011a;
Kuppel et al. 2018; Benettin and Bertuzzo 2018; Harman 2019; Wang et al. 2023). These studies have shown that most water
flowing to streams consists of a mixture of multiple ages, with TTDs spanning timescales from days to decades. The variability
in TTDs is influenced by changes across space and time (Klaus and McDonnell, 2013; Kirchner, 2016; Wang et al., 2025).
In recent years, studies focused on time-variable transit time distributions by applying the StorAge Selection (SAS) approach
(Botter et al., 2011; van der Velde et al., 2012; Hrachowitz et al., 2016; Harman, 2019) combined with catchment scale transport
models. The SAS formulation captures the age heterogeneity in hydrological systems by defining the relationship between the
distribution of ages stored in the hydrological system (residence time distribution, RTD) and the ages removed as outflows
(TTD). By applying SAS models with multiple functional forms, such as beta (van der Velde et al., 2015), gamma (Harman,
2019), and piecewise linear (Fenicia et al., 2006; McMillan et al., 2012) distributions, and tracking water fluxes, several





studies have highlighted the temporal variability of TTDs and demonstrated how transport processes can differ under varying conditions, such as between wet and dry periods (Benettin et al., 2015b; Harman, 2015; Kaandorp et al., 2018a). Moreover, using the SAS formulation and conceptualising the catchment as a multi-bucket system, studies have emphasised the partial

age mixing processes of recent precipitation contributing to different fluxes, including evapotranspiration (van der Velde et al., 2015; Maxwell et al., 2019), and macropore flow in the shallow subsurface (Hrachowitz et al., 2013; Klaus et al., 2013; Sprenger et al., 2016). This preferential flow of precipitation was found to become more prevalent with increasing soil moisture by bypassing smaller pore volumes and releasing younger water (Klaus et al., 2013; Wiekenkamp et al., 2016) and occasionally triggered by high precipitation intensities, leading to overland flow (Türk et al., 2024).

However, despite these findings of partial mixing in the unsaturated zone and the potential of preferential release of young water from groundwater, in many SAS applications the age composition of groundwater flow to the stream is often simplified by assuming uniform mixing of stored ages (e.g., Benettin et al. 2015a; Birkel et al. 2015; Ala-Aho et al. 2017; Knighton et al. 2019; Hrachowitz et al. 2013, 2021; Salmon-Monviola et al. 2025), noting that SAS functions are neither straightforward to measure nor parameterize. This simplification is typically adopted i) to maintain model simplicity, ii) due to the lack of

robust characterization of subsurface heterogeneity and its induced mixing mechanisms, and iii) due to the limited availability of detailed observations of groundwater flow processes, leaving gaps that must be filled by assumptions such as complete mixing of stored water ages. Nevertheless, several studies have emphasized that TTD estimates depend strongly on the chosen mixing assumptions in SAS models, thereby introducing uncertainty into estimates of transport timescales (van der Velde et al., 2012, 2015; Borriero et al., 2023). Reducing the complexity of groundwater storage representation by employing a

single, uniform SAS function shape may, therefore, oversimplify actual groundwater flow processes, potentially leading to erroneous conclusions in the estimation of water transit times.

Indeed, increasing evidence suggests that groundwater systems may not be completely mixed, and the preferential release of young water may be a ubiquitous feature of groundwater in heterogeneous aquifers (Berkowitz and Zehe, 2020; Hansen and Berkowitz, 2020a) for several reasons: (i) time-variant hydrological and climatic conditions (Maxwell et al., 2016), (ii)

generally low longitudinal and transversal dispersivities leading to little mixing, and (iii) complex structural heterogeneities influenced by geology, soil properties, and land use (Janos et al., 2018). Therefore, SAS functions should reflect that flow processes are transient and that groundwater contributes to the nonlinearity of flow processes and catchment responses (Kaandorp et al., 2018a).

Furthermore, instead of assuming a single mixed reservoir, and following the conceptualization of Zuber (1986), groundwater

is typically described by considering the mixing of active (water that contributes to flow) and passive groundwater storage volumes (water that mixes with the tracer signal of the active water volume but does not contribute directly to flow) (Fenicia et al., 2010; Birkel et al., 2011a; Hrachowitz et al., 2015). Birkel et al. (2011a) emphasised that the presence and extent of the passive storage can significantly influence the interpretation of tracer signals within a catchment. Yet, the extent to which the passive storage volumes and their associated mixing assumptions shape tracer signals, particularly when combined with

different SAS assumptions (e.g., complete mixing vs. partial mixing), still remains to some extent unknown. However, adopting more complex SAS parameterisations with additional parameters may exacerbate model uncertainty, particularly given the





limited availability of tracer data to constrain these parameters (Beven, 2006). Consequently, systematically testing different groundwater SAS shapes against long-term tracer observations is critical for assessing whether explicitly representing preferential groundwater flow (and associated SAS functions) meaningfully affects the quantification of transit time distributions in catchment-scale isotope-based transport models.

The main objective of this study was to evaluate how variations in SAS function parameterisations for preferential flow in the unsaturated root zone and groundwater influence estimated transit times and tracer composition at the stream outlet. By systematically comparing the effect of multiple mixing assumptions on the simulation of observed streamflow tracer data, we tested the hypothesis that preferential discharge of young groundwater can be significant and, therefore, should be represented by appropriate SAS functions. Additionally, we examined whether (and how) the extent and mixing assumptions of passive groundwater storage influence the interpretation of tracer signals and the estimation of transit times.

We specifically addressed the following research questions:

1. *Do precipitation and stream water tracer data have sufficient information content to identify and characterize preferential groundwater flow processes using different SAS function shapes, and if so, which SAS functions best represent these processes at the catchment scale?*

2. *Does explicitly accounting for preferential groundwater flow using different SAS functions significantly affect catchment-scale transit time distributions and the interpretation of tracer signals?*

3. *To what extent does the passive storage volume and associated mixing assumptions influence the representation of preferential groundwater flow, the estimated transit time distributions, and the interpretation of tracer signals at the catchment scale?*

To answer these questions, we used long-term hydrological and $\delta^2$H data from two contrasting headwater catchments. Each site exhibits distinct seasonal variability in runoff stable isotope signatures: one catchment displays minor isotopic variations during baseflow and sharp event-based responses (a "flashy" catchment), while the other catchment exhibits pronounced isotopic seasonality even during baseflow conditions. We implemented a time-variant TTD modelling framework capable of representing various mixing scenarios within these catchments.

## 2 Materials and methods

### 2.1 Study sites

The study sites for this study were the Hydrological Open Air Laboratory (HOAL) in Petzenkirchen, Lower Austria (Blöschl et al., 2016), and the Wüstebach headwater catchment in Germany's Eifel National Park (Bogena et al., 2014). The HOAL covers 66 hectares, and features a humid climate with a mean annual air temperature of around 9.5°C. The mean annual precipitation and runoff are approximately $823\,mm\,yr^{-1}$ and $195\,mm\,yr^{-1}$, respectively. The elevation ranges from 268 to 323 m a.s.l., with a mean slope of 8 %. Predominant soil types include Cambisols (57 %), Planosols (21 %), Kolluvisols (16



%), and Gleysols (6 %). The area's geology consists of Tertiary fine sediments of the Molasse underlain by fractured siltstone. Land use primarily includes agriculture (commonly maize, winter wheat, and rapeseed) (87 %), supplemented by forest (6 %),

pasture (5 %), and paved areas (2 %) (Blöschl et al., 2016).

The Wüstebach headwater catchment, part of the Lower Rhine/Eifel Observatory within the TERENO network, covers 38.5 hectares. It is characterized by a humid climate, with an annual temperature of around 7°C, mean annual precipitation of about $1200\,mm\ \mathrm{yr}^{-1}$, and mean annual runoff of $700\,mm\ \mathrm{yr}^{-1}$. The catchment's elevation ranges from 595 to 630 m a.s.l., with gentle hill slopes surrounding a relatively flat riparian area near the stream. The bedrock is primarily Devonian shales,

interspersed with sandstone inclusions and overlaid by periglacial layers. The hillslopes predominantly comprise Cambisols, while the riparian area features Gleysols and Histosols. The land use is primarily spruce forest (Bogena et al., 2018).

## 2.2 Hydrological and tracer data

We used daily hydro-meteorological data from October 2013 to 2019 for the HOAL catchment (Fig. 1a, 1b) and from October 2009 to October 2013, for the Wüstebach catchment (Fig. 1c, 1d). For the Wüstebach catchment, deforestation in October 2013

led to changes in catchment flow generation processes (Hrachowitz et al., 2021). Therefore, the period after deforestation was not used for the analyses.

In the HOAL, precipitation data were recorded using a weighing rain gauge located $200\,m$ from the catchment outlet, and stream discharge was measured at the catchment outlet using a calibrated H-flume. The precipitation samples for isotopic analysis were collected using an adapted Manning S-4040 automatic sampler located approximately $300\,m$ south of the catchment.

In addition to precipitation samples, weekly grab samples of streamflow were collected at the catchment outlet for isotopic analysis. Additionally, event-based streamflow samples were collected using an automatic sampler, with the frequency of sampling adjusted based on flow rate thresholds (without exceeding sampling bottle capacity). Isotopic measurements of $\delta^{18}O$ and $\delta^2H$ were conducted using cavity ring-down spectroscopy (Picarro L2130-i and L2140-i), with an analytical uncertainty of $\pm 0.1\,‰$ for $\delta^{18}O$ and $\pm 1.0\,‰$ for $\delta^2H$.

In the Wüstebach catchment, precipitation data were obtained from a nearby meteorological station operated by the German Weather Service (Deutscher Wetterdienst, DWD station 3339), and stream discharge was measured using a V-notch weir for low flows and a Parshall flume for high flows (Bogena et al., 2014). The precipitation samples for isotopic analysis were collected at the Schöneseiffen meteorological station, located approximately 3 km northeast of the catchment at an elevation of 620 m a.s.l. Starting in June 2009, weekly precipitation samples were collected using a cooled storage rain gauge with 2.3-L HDPE

bottles (Stockinger et al., 2014). From September 2012 onward, the sampling resolution was increased to daily intervals (Fig. 1d) using a cooled automated sampler (Eigenbrodt GmbH & Co. KG, Germany; 250 mL PE bottles). Stream water samples for isotopic analysis were collected weekly at the catchment outlet as grab samples. Cavity ring-down spectroscopy (Picarro L2120-i, L2130-i) was used for water isotope analyses, with an analytical uncertainty of $\pm 0.1\,‰$ for $\delta^{18}O$ and $\pm 1.0\,‰$ for $\delta^2H$. All isotopic measurements are reported as per mil (‰) relative to Vienna Standard Mean Ocean Water (VSMOW).





**Figure 1.** Hydrological and tracer data of the HOAL and Wüstebach catchments. (a, c) daily measured streamflow $Q$ $(mmd^{-1})$ and precipitation $P$ $(mmd^{-1})$, (b, d) precipitation $\delta^2$H signals (light blue) and streamflow $\delta^2$H signals (dark blue); the size of the dots indicates the relative precipitation volume. For the HOAL catchment, the $\delta^2$H data of streamflow was further shown as the weekly grab samples (b, dark blue dots) and event samples (b, orange dots). For the HOAL catchment, precipitation $\delta^2$H samples are in daily resolution, whereas for the Wüstebach catchment, beginning in September 2012, the sampling frequency for precipitation $\delta^2$H increased from weekly to daily (d).





## 2.3 Hydrological model and tracer transport model

We used a process-based hydrological and transport model (Türk et al., 2024) based on the DYNAMITE modelling framework (Hrachowitz et al., 2014). Briefly, both the HOAL and Wüstebach catchments are conceptualised through five interconnected reservoirs: snow, canopy interception, unsaturated root zone, fast response storage, and groundwater with active and passive components (Fig. S 1). To route $\delta^2$H fluxes through the model, the storage-age selection function (SAS) approach (Rinaldo et al., 2015; Harman, 2015) was integrated into the hydrological model. This model formulation allows the simulation of water fluxes and tracer dynamics simultaneously, enabling the estimation of TTDs from the age distributions of stored water. The SAS function was formulated as the likelihood of selecting water parcels of different ages from catchment storage compartments (e.g., unsaturated zone) to outputs (e.g., streamflow or evapotranspiration). Further details on the model architecture and assumptions can be found in previous studies (Hrachowitz et al., 2014; Fovet et al., 2015). The water balance and flux equations for the two catchments in this study application are described in Türk et al. (2024).

Similar to previous tracer transport studies for the HOAL (Türk et al., 2024) and Wüstebach (Hrachowitz et al., 2021) catchments, we used beta distributions to formulate the SAS functions. Beta distributions are defined by two shape parameters ($\alpha$ and $\beta$). For all modelled SAS functions, except those representing preferential flow from the unsaturated root zone, the $\alpha$ and $\beta$ parameters were initially fixed at 1. This ensured uniform sampling of water parcels of different ages from catchment storage compartments into the outflows.

In the Wüstebach catchment, previous studies (Wiekenkamp et al., 2016; Hrachowitz et al., 2021) identified catchment soil wetness as the primary driver for activating preferential flow pathways in the unsaturated zone. Therefore, the SAS function representing preferential flow from the unsaturated root zone ($R_f$, Fig. S 1) was formulated as a time-variable function of soil wetness to reflect changes in transport processes between wet and dry soil conditions. The temporal variability in the SAS function was implemented through a time-dependent shape parameter $\alpha(t)$ (Eq. 2).

In the HOAL catchment, previous studies highlighted the non-linearity of preferential flow generation, where both precipitation intensity and soil moisture influence the activation of preferential flow pathways (Türk et al., 2024; Széles et al., 2020). To represent this behaviour, the SAS function was formulated with a time-variable shape parameter $\alpha(t)$. Here, $\alpha(t)$ varied as a function of soil moisture, and as a function of precipitation intensity ($P_I$, mm d$^{-1}$).

For the HOAL catchment time variability of $\alpha$ for preferential flow in the unsaturated root zone was defined as:

$$\alpha(t) = \begin{cases} \alpha_0, & \text{if } P_r(t) \geq P_{\text{thresh}} \\ 1 - \frac{S_r(t)}{S_{r,\max}}(1-\alpha_0), & \text{if } P_r(t) < P_{\text{thresh}} \end{cases} \qquad (1)$$

For the Wustebach catchment, time variability of $\alpha$ for the preferential flow in the unsaturated root zone was defined as:

$$\alpha(t) = 1 - \left(\frac{S_r(t)}{S_{r,\max}}\right)(1-\alpha_0) \qquad (2)$$

In both Equations 1 and 2, the shape parameter $\alpha$ controls the preferential release of younger water: values of $0 < \alpha < 1$ indicate a bias towards younger water parcels, whereas $\alpha = 1$ corresponds to uniform sampling. The $\alpha_0$ is a calibration parameter





representing the lower bound between 0 and 1, allowing $\alpha(t)$ to vary between $\alpha_0$ and 1. When soil moisture is low ($S_r(t) \ll S_{r,\text{max}}$), $\alpha(t)$ approaches 1, indicating uniform sampling. As soil moisture increases ($S_r(t)$ approaches $S_{r,\text{max}}$), $\alpha(t)$ decreases towards $\alpha_0$, reflecting a stronger preference for younger water. In Equation 1, the lower bound $\alpha_0$ is applied directly whenever precipitation intensity exceeds a certain threshold ($P_{\text{thresh}}$).

### 2.3.1 Model calibration and evaluation

We used daily time steps in the model parameter calibration for the period from October 2014 to 2019 for the HOAL catchment and for the period from October 2010 to October 2013 for the Wüstebach catchment to simulate streamflow $Q$ ($mm\,\text{d}^{-1}$) and $\delta^2$H signature. The model warm-up period was one year for both catchments; i.e., from October 2013 to October 2014 for the HOAL catchment, and from October 2009 to October 2010 for the Wüstebach catchment.

For model parameter optimization, we used the Differential Evolution algorithm (Storn and Price, 1997) and an objective function that combined five performance criteria related to streamflow and $\delta^2$H dynamics. The objective function included the Nash-Sutcliffe efficiencies (NSE) of streamflow, logarithmic streamflow, the flow duration curve, the runoff coefficient averaged over three months, and the NSE of the $\delta^2$H signal in streamflow (Table S 2). These individual performance metrics were aggregated into the Euclidean distance $D_E$ to the perfect model, with equal weights assigned to streamflow and the $\delta^2$H signature, according to:

$$D_E = \sqrt{\frac{1}{2}\left(\frac{\sum_{i=m}^{M}(1-E_{Q,m})^2}{M} + \frac{\sum_{i=n}^{N}(1-E_{18O,n})^2}{N}\right)} \tag{3}$$

Where $M = 4$ is the number of performance metrics with respect to streamflow, $N = 1$ is the number of performance metrics for tracers in each combination, and $E$ is the evaluation matrix based on goodness-of-fit criteria. The Euclidean distance $D_E$ to the "perfect model" (where $D_E = 0$ indicates a perfect fit) was used to ensure that overall model performance remained balanced. Only solutions achieving $D_E \leq 1$ were accepted as feasible solutions for further analysis. The accepted solutions were then ranked in order of decreasing $D_E$, and the solution with the lowest $D_E$ was selected as the optimal parameter set for TTD estimations.

### 2.3.2 Sensitivity test of root zone and groundwater SAS functions

To evaluate whether precipitation and stream water tracer data carry sufficient information to identify preferential flow paths, and to examine how different groundwater SAS function shapes might affect catchment-scale transit time distributions, we conducted a stepwise analysis (Fig. 2). In this approach, the model was run with calibrated hydrological parameters with different configurations for the StorAge Selection (SAS) function shape parameter $\alpha_0$ in the unsaturated root zone and $\alpha$ the groundwater compartments. To identify if variations in transit time distributions were solely due to differences in age selection formulated by StorAge Selection (SAS), we kept all other hydrological parameters constant across scenarios. By using the same calibrated parameters (e.g., maximum percolation rate, storage capacities, and flow path configurations) for HOAL and



Wüstebach in each scenario, any differences in the simulated age distributions can then be attributed to the changes in the SAS function formulation. We first tested the sensitivity of the $\delta^2$H signal to changes in the root zone's preferential flow by setting lower bound of SAS shape parameter $\alpha_0$ to 0.1 (very young-water preference), 0.7 (young-water preference), 1.0 (uniform selection), and 5.0 (older-water preference), while keeping the groundwater SAS function uniform (i.e. $\alpha = 1$; Fig. 2a). This

220  approach assesses whether root-zone preferential pathways alone could reveal a strong impact on the streamflow $\delta^2$H time series and the inferred transit times. Next, we tested the sensitivity of the $\delta^2$H signal to changes in the groundwater SAS function (Fig. 2b) by varying $\alpha$ across the same range—0.1, 0.7, 1.0, and 5.0—while fixing the previously calibrated optimized $\alpha_0$ value for the root zone. This second test was designed to show if (and how) preferential groundwater flow influences transit time distributions and tracer simulations. At each step, we evaluated the model's performance in simulating $\delta^2$H using

225  Spearman rank correlation, $\text{NSE}_{\delta^2\text{H}}$, and $\text{MAE}_{\delta^2\text{H}}$. We then calculated daily cumulative TTDs and compared how the mean of these cumulative TTDs changed across all scenarios.



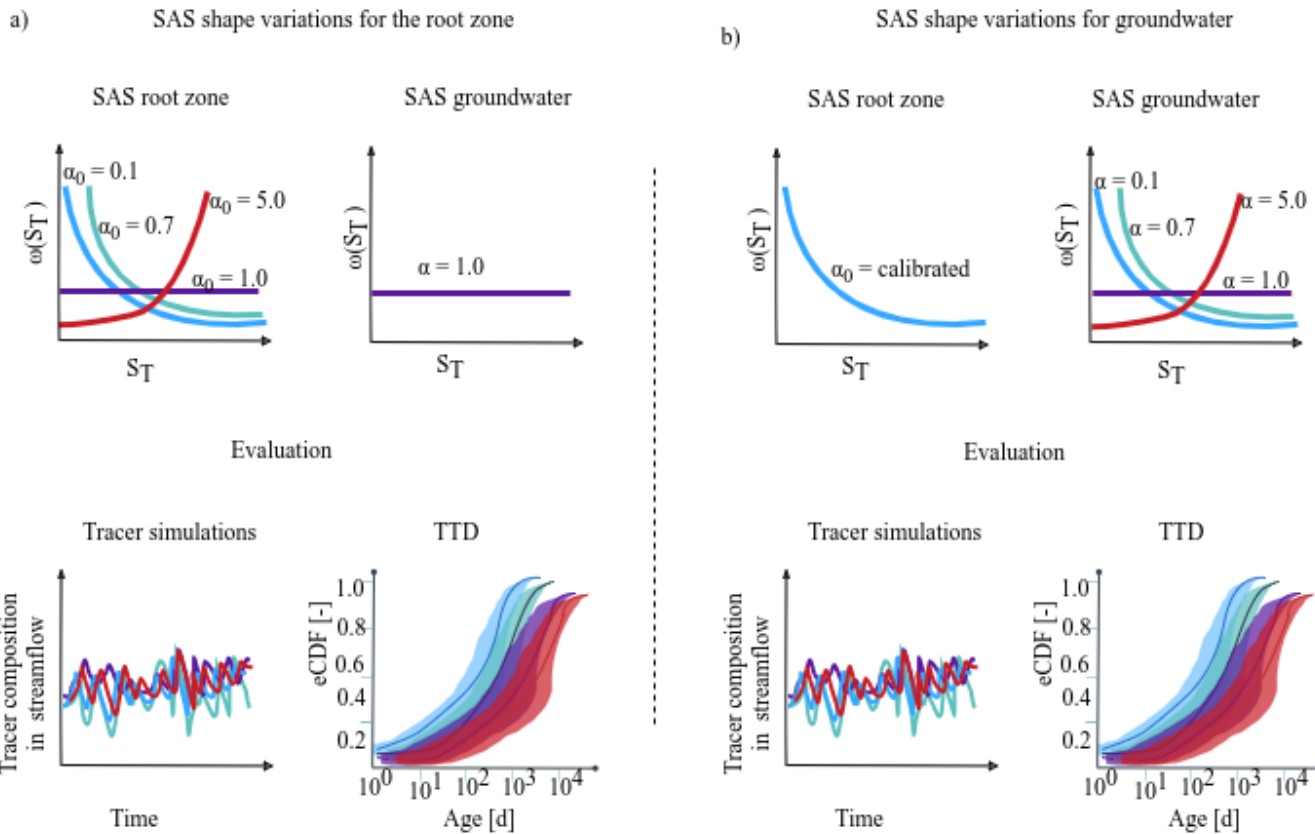

**Figure 2.** Conceptual representation of the stepwise analysis, showing how different SAS functions, formulated with shape parameter lower bound $\alpha_0$ in the root zone preferential flow and ($\alpha$) for groundwater, flow impact the simulated tracer signals and inferred TTDs. (a, b) The top plots illustrate how the SAS function varies with $\alpha$. Values of $\alpha < 1$ indicate a stronger preference for younger water (blue line), whereas $\alpha > 1$ implies preferential release of older water (red line). The x-axis, $S_T$, represents the age-ranked storage, and the y-axis, $\omega(S_T)$, is the relative probability of releasing water of that age. The lower plots illustrate the modelled tracer time series and the resulting cumulative transit time distribution. (a) The root zone $\alpha_0$ is varied from 0.1 (very young water preference) to 5.0 (older-water preference) while the groundwater compartment remains uniform; (b) the root zone SAS function $\alpha_0$ is assigned to its calibrated value, and the groundwater $\alpha$ is varied from 0.1 (very young water preference) to 5.0 (older-water preference).



## 2.4 Passive groundwater storage volumes and mixing assumptions with the active groundwater storage

To test whether and to what extent the mixing of the passive groundwater storage with the active groundwater modulates the $\delta^2$H signal in streamflow—and, consequently, influences model performance and inferred transit times—we extended the stepwise analysis (Fig. 2b) by varying passive storage volumes (Fig. 3). We used the calibrated SAS function shape parameter ($\alpha_0$) for the root zone preferential flow in combination with four different groundwater mixing scenarios: a strong preference for younger water ($\alpha = 0.1$), a preference for younger water ($\alpha = 0.7$), uniform selection ($\alpha = 1.0$), and a preference for older water ($\alpha = 5.0$) (Fig. 3 a). These scenarios were each applied to three different passive storage volumes $S_{s,p}$ = 500 $mm$, 1000 $mm$, and 5000 $mm$. We evaluated model performance in simulating streamflow $\delta^2$H by comparing measured and modeled isotope signals using NSE$_{\delta^2\text{H}}$ and MAE$_{\delta^2\text{H}}$. To further compare the inferred transit time distributions (TTDs) in each scenario (Fig. 3b), we calculated daily cumulative TTDs and assessed how the mean of these distributions changed as passive storage volumes and mixing assumptions varied.





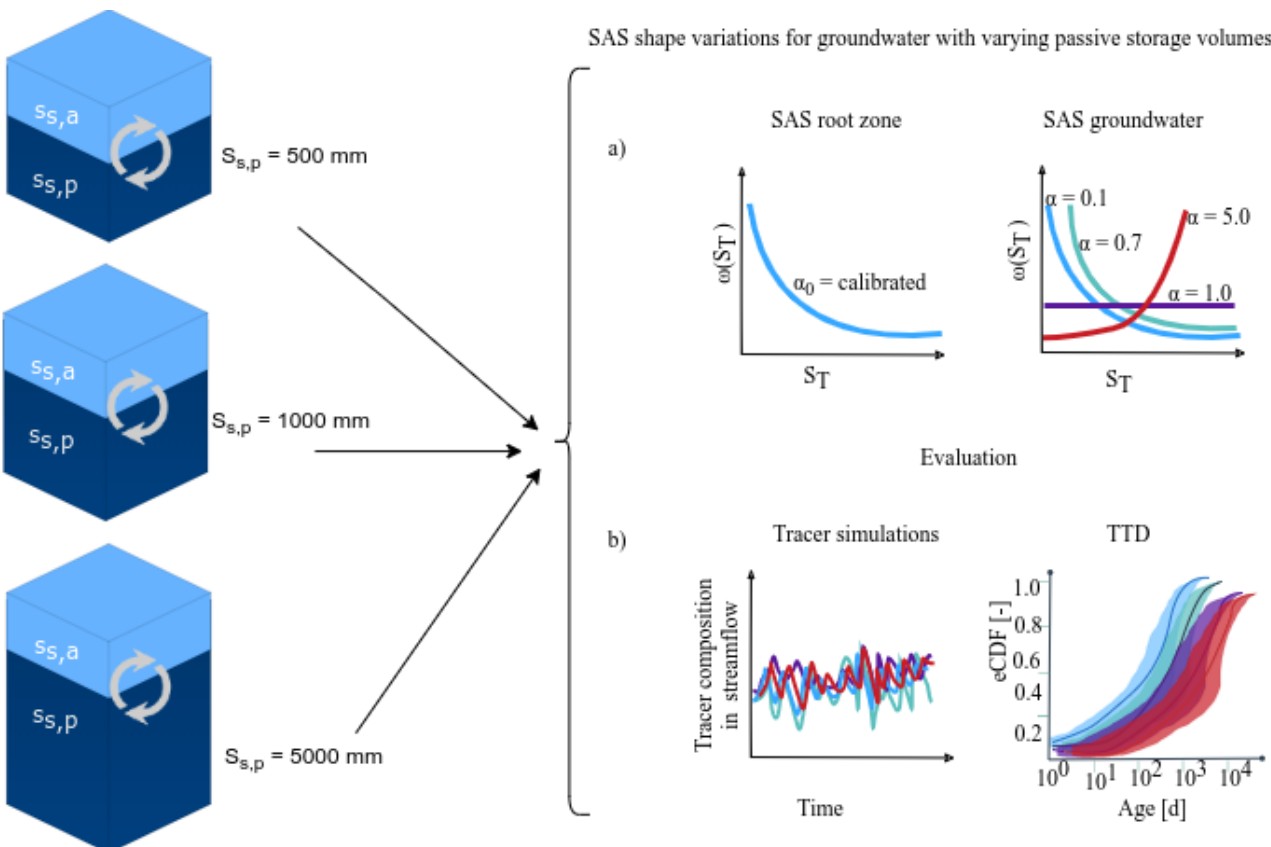

**Figure 3.** Comparative analysis illustrating how different passive storage volumes ($S_{s,p}$: 500 $mm$, 1000 $mm$, 5000 $mm$) mix with the active storage volume ($S_{s,a}$) under various SAS function shapes for groundwater. (a) The root-zone SAS function $\alpha_0$ is set to its calibrated value, and the groundwater $\alpha$ is varied from 0.1 (very young-water preference) to 5.0 (older-water preference). Values of $\alpha < 1$ indicate a stronger preference for younger water (blue line), whereas $\alpha > 1$ implies preferential release of older water (red line). The x-axis, $S_T$, represents the age-ranked storage, and the y-axis, $\omega(S_T)$, is the relative probability of releasing water of that age, (b) shows the modelled tracer time series and the resulting cumulative transit time distribution.




## 3   Results

### 3.1   Variation of $\delta^2$H in precipitation and streamflow

In the HOAL catchment, $\delta^2$H values in precipitation ranged from $-3.0$ ‰ to $-150.0$ ‰ (Fig. 1b), with a volume-weighted mean of $-67.7$ ‰ $\pm 31.9$ ‰. Event-based streamflow $\delta^2$H samples ranged from $-26.2$ ‰ to $-108.0$ ‰ (Fig. 1b), while weekly streamflow $\delta^2$H samples ranged from $-73.2$ ‰ to $-75.2$‰. The volume-weighted mean of stream samples was $-71.6$ ‰ $\pm 6.1$ ‰.

In the Wüstebach catchment, $\delta^2$H values in precipitation ranged from $-4.3$ ‰ to $-163.2$ ‰ (Fig. 1d, light blue dots), with a volume-weighted mean of $-52.2$ ‰ $\pm 21.4$ ‰. Weekly streamflow $\delta^2$H values exhibited smaller variations, ranging from $-45.6$ ‰ to $-57.1$ ‰ (Fig. 1d). The volume-weighted mean of stream samples was $-53.2$ ‰ $\pm 1.4$ ‰.

### 3.2   Model calibration

Model calibration resulted in 55 feasible parameter solutions for HOAL (Fig. S 2) and 190 feasible parameter solutions for the Wüstebach catchment (Fig. S 3). The model reproduced the main features of the hydrograph and captured both the timing and magnitude of high and low flow events for the simulation period from October 2014 to 2019 for HOAL (Fig. S 4a, d) and from October 2010 to October 2013 for the Wüstebach catchment (Fig. S 4e, h).

For the HOAL catchment, the mean Nash-Sutcliffe efficiency of streamflow ($\text{NSE}_Q$) for the 55 solutions was 0.60 (Fig. S 5). Minor dissimilarities occurred during the spring of 2016, when low flows were overestimated (Fig. S 4a). Nevertheless, the model simulated most other observed flow signatures reasonably well (Fig. S 5). Among the 55 solutions, the mean NSE for low flows ($\text{NSE}_{\log Q}$) was 0.65, for the flow duration curve ($\text{NSE}_{\text{FDC}}$) was 0.53, and for the three-month averaged runoff ratio ($\text{NSE}_{\text{RC}}$) it was 0.85. For several rain events, the model captured $\delta^2$H fluctuations during high flows and maintained a stable $\delta^2$H signal during low flows, with a mean $\text{NSE}_{\delta^2\text{H}}$ of 0.51. Overall, the Euclidean distance ($D_E$) for these 55 solutions ranged from 0.60 to 0.33 (Fig. S 5).

For the Wüstebach catchment, the mean Nash-Sutcliffe efficiency of streamflow ($\text{NSE}_Q$) for the 190 solutions was 0.78 (Fig. S 5). Minor dissimilarities occurred during the spring of 2012, when low flows were overestimated, and winter of 2012 when peak flows were underestimated (Fig. S 4e). Among the 190 solutions, the mean NSE for low flows ($\text{NSE}_{\log Q}$) was 0.65, for the flow duration curve ($\text{NSE}_{\text{FDC}}$) it was 0.93, and for the three-month averaged runoff ratio ($\text{NSE}_{\text{RC}}$) it was 0.91. For several rain events, the model captured $\delta^2$H fluctuations during high flows and maintained a stable $\delta^2$H signal during low flows, with a mean $\text{NSE}_{\delta^2\text{H}}$ of 0.58. Overall, the Euclidean distance ($D_E$) for these 190 solutions ranged from 0.62 to 0.32 (Fig. S 5).

### 3.3   Catchment transit times

In the HOAL catchment, the fraction of streamflow younger than 1000 days exhibited considerable variability, ranging from 5 % to 50 % (Fig. 4a). The mean fraction of discharge younger than 1000 days was 13 %; it increased to 15 % during wet



periods and decreased to 10 % during dry periods (Fig. 4a). The value of the fraction of streamflow younger than 90 days,

270    $F_Q(T < 90\,\text{days})$, varied widely within the same month, ranging from 2 % to 45 %; however, the mean $F_Q(T < 90\,\text{days})$ across months did not exhibit pronounced seasonal patterns (Fig. 4b). The mean value of simulated relative soil saturation $(S_r/S_{r,\max})$ varied from 0.25 to 0.60 (Fig. 4c).

In the Wüstebach catchment, the mean fraction of discharge younger than 1000 days was 27 %, increasing to 35 % during wet periods and decreasing to 20 % during dry periods (Fig. 4d). The value of the fraction of streamflow younger than 90 days,

275    $F_Q(T < 90\,\text{days})$ within the same month between 5 % and 30 % (Fig. 4e), with mean values exhibiting seasonal patterns. The monthly mean of simulated relative soil saturation $(S_r/S_{r,\max})$ ranged from approximately 0.60 to 0.98 (Fig. 4f).





**Figure 4.** Modelled empirical cumulative transit time distributions (TTDs) for daily streamflow in the (a) HOAL and (d) Wüstebach catchments. The colour of the lines corresponds to the wetness state, where dark blue indicates a wet period and dark red indicates a dry period. In panels (a) and (d), the mean of the empirical cumulative TTDs is shown for the entire tracking period (black line), the dry period (red line), and the wet period (dark blue line). The fraction of streamflow younger than 90 days, $F_Q(T < 90)$, grouped by month of the year, is shown in panels (b) and (e) for the HOAL and Wüstebach catchments, respectively. Simulated relative soil wetness $\left(\frac{S_r}{S_{r,\max}}\right)$, also grouped by month of the year, is shown in panels (c) and (f) for the HOAL and Wüstebach catchments, respectively. (b, c, e, f) Green triangles indicate the mean values.





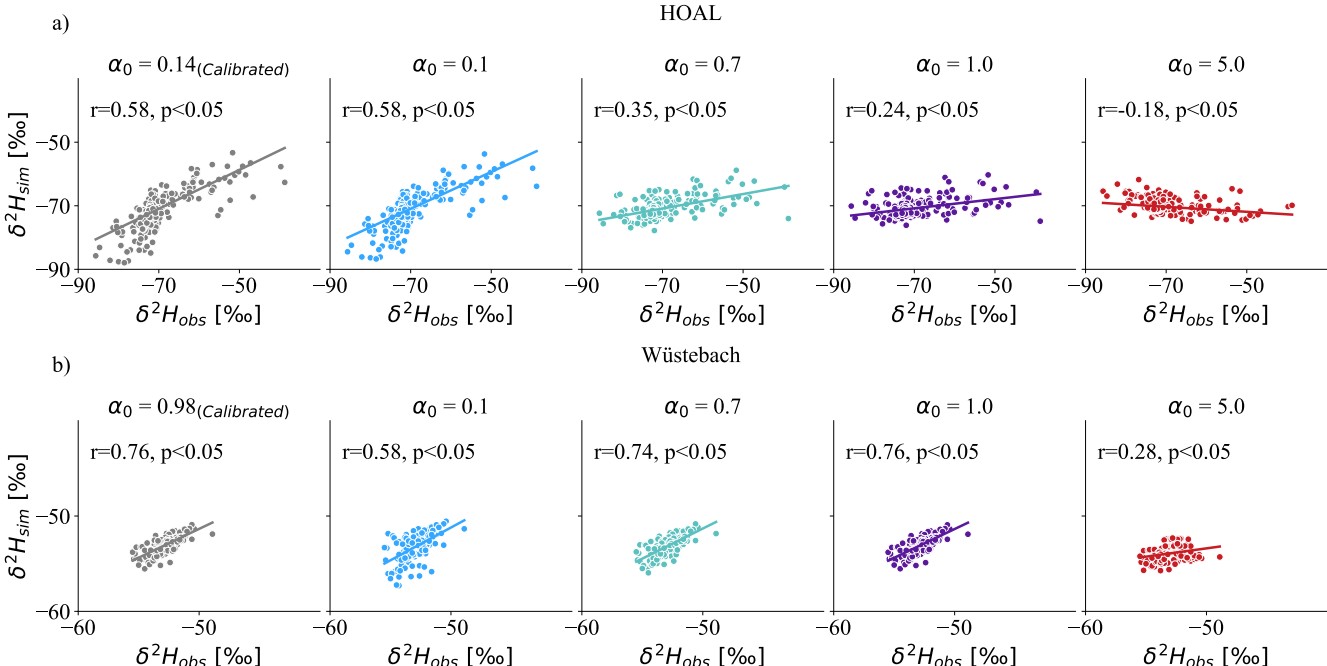

**Figure 5.** Spearman rank correlations between simulated (y-axis) and observed (x-axis) $\delta^2$H signals in streamflow based on varying the SAS shape parameter $\alpha$ [-] in the root zone for (a) HOAL and (b) Wüstebach. The simulations range from very young preference ($\alpha = 0.1$) to old water preference ($\alpha = 5$) for the unsaturated root zone preferential flow, while the groundwater flow was uniformly sampled ($\alpha = 1$).

## 3.4 Sensitivity of $\delta^2$H simulations and TTD estimation to different SAS functions in the root zone

In the HOAL catchment, the calibrated root-zone SAS shape parameter lower bound $\alpha_0 = 0.14$ indicated a strong preference for very young water through unsaturated root zone preferential flow pathways, suggesting that precipitation rapidly reached the stream with minimal mixing with stored water. In the Wüstebach catchment, the calibrated SAS shape parameter lower bound $\alpha_0 = 0.98$ suggested a slight preference for young water in the root zone's preferential flows.

For both catchments, root-zone preferential flow SAS functions ranging from a strong young water preference ($\alpha_0 = 0.1$) to uniform sampling ($\alpha_0 = 1.0$) produced high (positive) Spearman rank correlations ($r$) between modeled and observed $\delta^2$H. In contrast, an old-water preference ($\alpha_0 = 5.0$) yielded negative or weak correlations, indicating a poor fit to the observed tracer signals. In HOAL, the $r$ values ranged between 0.58, and $-0.18$ for values of $\alpha_0$ between d 0.1 and 5.0 (Fig. 5a). The corresponding Nash–Sutcliffe efficiencies (NSE$_{\delta^2\text{H}}$) ranged between 0.56, and $-0.25$ (Table 1). In Wüstebach, the $r$ values for simulated $\delta^2$H ranged between 0.58, and 0.28 for values of $\alpha_0$ between d 0.1 and 5.0 (Fig. 5b). The corresponding NSE$_{\delta^2\text{H}}$ ranged beetwen $-0.14$, and 0.51 (Table 1).

For both catchments, root-zone preferential flow SAS functions from a preference for young water ($\alpha_0 = 0.1$) to old water ($\alpha_0 = 5.0$) influenced the TTD for ages up to 300 days ($T < 300$). This was due to the fact that root-zone storage residence time



remained predominantly younger than 300 days (Fig. S 6a, c). Consequently, increasing $\alpha_0$ from $0.1$ to $5.0$ and thus reducing the relative contribution of younger flows (Fig. 6a, b), shifted the empirical cumulative distribution functions (eCDFs) toward older water within the first $300$ days. In the HOAL, the mean fraction of streamflow with $T < 300$ days reached about $10\,\%$ (Fig. 6a,) for all root-zone SAS formulations, whereas in Wüstebach, it was about $20\,\%$ (Fig. 6b). Overall, these results indicated that

295   root-zone SAS functions with young-water preferences improved the fit to observed streamflow isotopes, highlighting the importance of preferential flow pathways in shaping short transit times and streams $\delta^2\mathrm{H}$ interpretations.

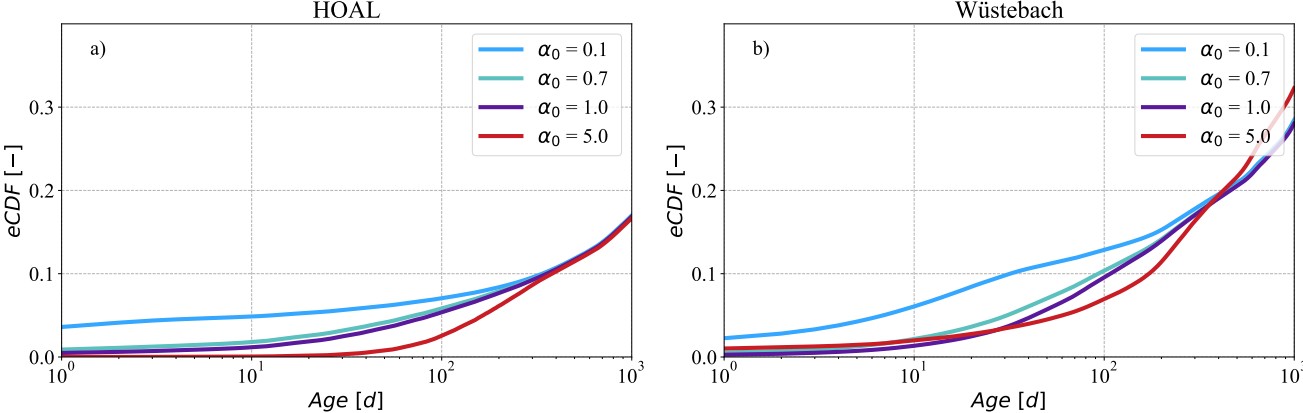

**Figure 6.** The mean of empirical cumulative distribution functions (eCDFs) of simulated transit times of daily discharge for the (a) HOAL and (b) Wüstebach catchments under varying SAS shape parameters in the unsaturated root zone ($\alpha_0 = 0.1, 0.7, 1.0, 5.0$). (a,b) The simulations range from very young preference ($\alpha_0 = 0.1$) to old water preference ($\alpha_0 = 5$) for the unsaturated root zone preferential flow, while the groundwater flow was uniformly sampled ($\alpha = 1$).





**Table 1.** Performance metrics for $\delta^2$H simulation results under various SAS parameter scenarios for the HOAL and Wüstebach catchments. The table includes the Nash- Sutcliffe efficiency ($NSE_{\delta^2 H}$) and mean absolute error ($MAE_{\delta^2 H}$), and Spearman rank correlation coefficients ($r_{\delta^2 H}$) based on SAS shape parameters ($\alpha_0$) variations in the root zone and groundwater SAS shape parameters ($\alpha$). Scenarios tested represent preferences for very young water ($\alpha = 0.1$), young water ($\alpha = 0.7$), uniform selection ($\alpha = 1.0$), and old water ($\alpha = 5.0$). For simulations testing SAS function variations in the root zone, the groundwater SAS function was kept uniform. Conversely, when testing groundwater SAS function variations, the root zone compartment was assigned its calibrated shape factor ($\alpha_0 = 0.14$ for HOAL and $\alpha_0 = 0.98$ for Wüstebach).

| Catchment | Metric | SAS Variation: Root Zone | | | | SAS Variation: Groundwater | | | |
|---|---|---|---|---|---|---|---|---|---|
| | | $\alpha_0 = 0.1$ | $\alpha_0 = 0.7$ | $\alpha_0 = 1.0$ | $\alpha_0 = 5.0$ | $\alpha = 0.1$ | $\alpha = 0.7$ | $\alpha = 1.0$ | $\alpha = 5.0$ |
| HOAL | $NSE_{\delta^2 H}$ | 0.56 | 0.28 | 0.15 | -0.25 | -0.83 | 0.55 | 0.56 | 0.55 |
| | $MAE_{\delta^2 H}$ | 2.46 | 2.85 | 3.06 | 4.02 | 4.75 | 2.54 | 2.48 | 2.48 |
| | $r_{\delta^2 H}$ | 0.55 | 0.35 | 0.24 | -0.18 | 0.54 | 0.55 | 0.56 | 0.56 |
| Wüstebach | $NSE_{\delta^2 H}$ | -0.14 | 0.47 | 0.51 | -0.81 | 0.10 | 0.05 | 0.51 | 0.19 |
| | $MAE_{\delta^2 H}$ | 0.90 | 0.64 | 0.61 | 1.14 | 0.74 | 0.91 | 0.61 | 0.83 |
| | $r_{\delta^2 H}$ | 0.58 | 0.74 | 0.76 | 0.28 | 0.74 | 0.71 | 0.76 | 0.75 |

## 3.5 Sensitivity of $\delta^2$H simulation and TTD estimation to different SAS functions for groundwater

The Spearman rank correlation coefficients ($r$) between simulated and observed $\delta^2$H signals in streamflow, obtained by varying the SAS shape parameter $\alpha$ in groundwater, are shown in Figure 7. For the HOAL catchment, $r$ values ranged from 0.54 to 0.60, indicating that, in contrast to the root-zone, changes in the groundwater SAS function had minimal impact on the fit between simulated and observed $\delta^2$H signals (Fig. 7a). In the Wüstebach catchment, $r$ values only slightly increased from 0.71 ($\alpha = 0.1$) to 0.76 ($\alpha = 1.0$) before decreasing slightly at $\alpha = 5.0$ to 0.75. In both catchments, the correlations remained consistently strong across all $\alpha$ values tested (Fig. 7a, b).

A stronger preference for young water ($\alpha = 0.1$) led to approximately 25 % of streamflow being younger than 1000 days in the HOAL (Fig. 9a) and 35 % in the Wüstebach (Fig. 9b). In contrast, an older-water preference ($\alpha = 5.0$) shifted the distribution and reduced the proportion of streamflow being younger than 1000 days around to 5% in the HOAL and to 12% in the Wüstebach. This shift, resulting from changing the SAS function parameter $\alpha$ from 0.1 to 5.0, produced a variability of approximately 20 % in HOAL and 23 % in Wüstebach in the proportion of streamflow composed of water younger than 1000 days (Fig. 9a, b).





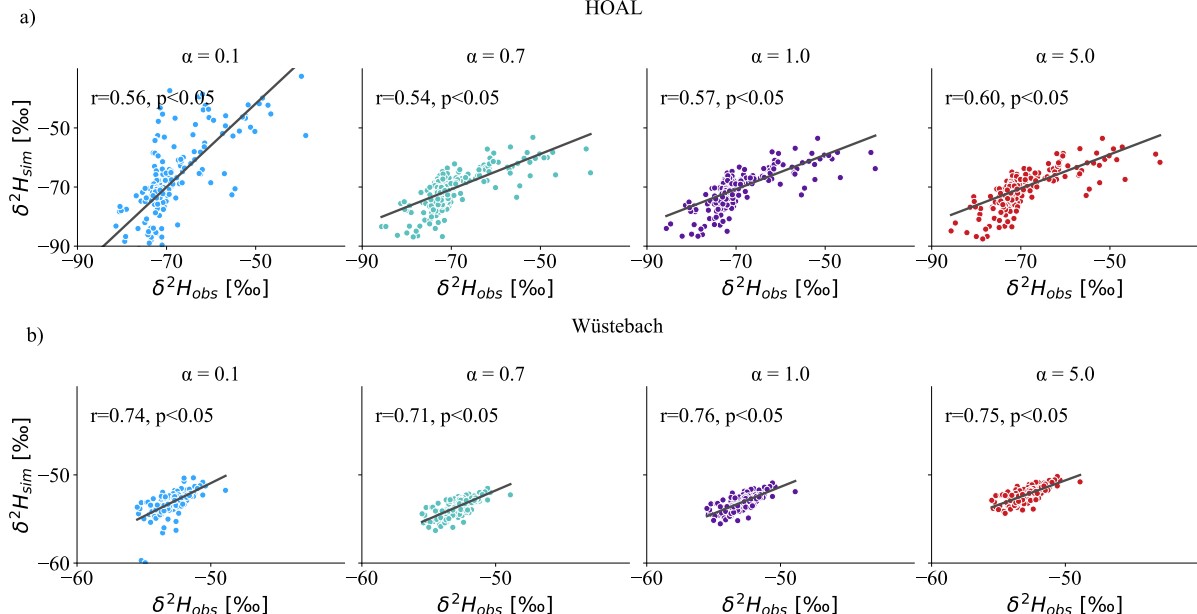

**Figure 7.** Spearman rank correlations between simulated (y-axis) and observed (x-axis) $\delta^2$H signals in streamflow based on varying the SAS shape parameter $\alpha$[-] in groundwater for (a) HOAL and (b) Wüstebach. The simulations ranged from very young water preference ($\alpha = 0.1$) to old water preference ($\alpha = 5$) for the groundwater, while for the root zone compartment, a calibrated value was used ($\alpha_0 = 0.14$ for HOAL and 0.98 for Wüstebach).





**Figure 8.** Simulation of $\delta^2$H in streamflow based on varying SAS shape parameter $\alpha$ [-] in groundwater for (a) HOAL, 2015 and (b) Wüstebach, 2011. Simulations over the full tracking period are provided in Figure S 7. The simulations ranged from very young water preference ($\alpha = 0.1$) to old water preference ($\alpha = 5$) for the groundwater, while for the root zone compartment, a calibrated value was used ($\alpha_0 = 0.14$, for HOAL and 0.98 for Wüstebach). The simulated $\delta^2 H$ signals from the model are illustrated with blue, turquoise, purple, and red lines corresponding to $\alpha$ values of 0.1, 0.7, 1.0, and 5.0, respectively. The grey-shaded area shows the measured streamflow (Q, mm d$^{-1}$) for both catchments.





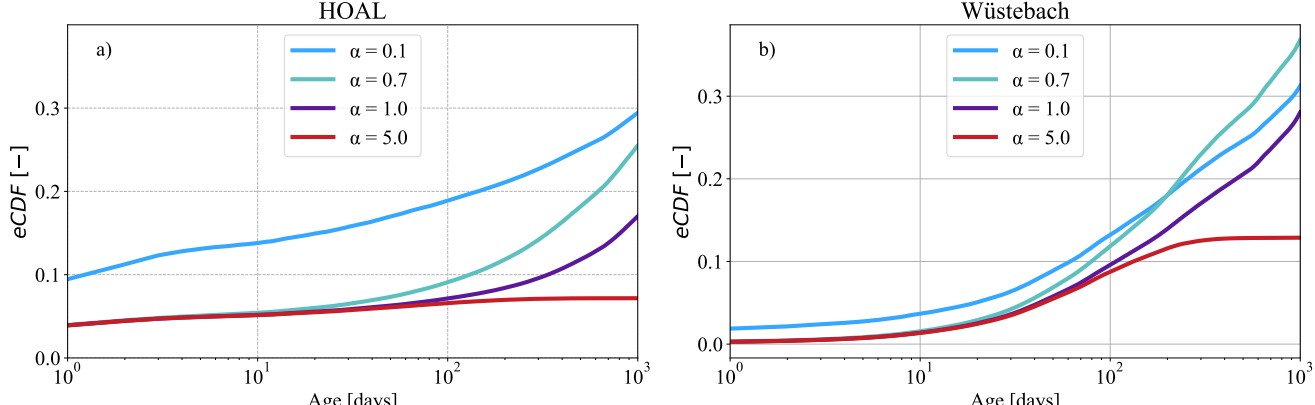

**Figure 9.** The mean of empirical cumulative distribution functions (eCDFs) of simulated transit times of daily discharge for the (a) HOAL and (b) Wüstebach catchments under varying SAS shape parameters ($\alpha = 0.1, 0.7, 1.0, 5.0$) for groundwater. Lower $\alpha$ values favour younger water, producing younger transit time distributions, while higher $\alpha$ values shift the distribution toward older water. The mean of inferred TTD lines are illustrated with blue, turquoise, purple, and red lines corresponding to $\alpha$ values of 0.1, 0.7, 1.0, and 5.0, respectively. (a, b) The simulations ranged from very young water preference ($\alpha = 0.1$) to old water preference ($\alpha = 5$ ) for the groundwater, while for the root zone compartment, a calibrated value was used ($\alpha_0 = 0.14$, for HOAL and 0.98 for Wüstebach).

### 3.6 Variation in the streamflow tracer signal under different passive storage volumes and mixing assumptions

The results addressing the extent to which passive storage volume and associated mixing assumptions influence the representation of preferential groundwater flow, the estimated transit time distributions, and the interpretation of tracer signals at the catchment scale, are presented in Figures 10 and 11. Briefly, and somewhat surprisingly, the findings show that increasing the passive storage volume dampens the contribution of young water, shifts the overall transit time distribution towards older ages, and reduces variability in the $\delta^2$H signal.

Simulations with varying $S_{S,p}$ volumes and different mixing assumptions (Figs. 10, 11) resulted in distinct $\delta^2$H responses in streamflow. In both catchments, an active storage volume rate equivalent to approximately 1 % of the passive storage volume was needed to attenuate the simulated tracer signal in line with observations. In the HOAL, a passive storage volume of $S_{S,p} = 500$ mm (Fig. S 8a) was sufficient to achieve this, while in Wüstebach, a much larger volume of $S_{S,p} = 5000$ mm was necessary (Fig. S 8 b). SAS shape parameters indicating a young-water preference ($\alpha = 0.1$) resulted in variable $\delta^2$H signals in streamflow, whereas an older-water preference ($\alpha = 5.0$) led to stronger dampening (Figs. 10, 11). Once the volume rate between active and passive storage fell below 1 %, further increases in $S_{S,p}$ had little effect on model performance (Table 2). The Nash–Sutcliffe Efficiency (NSE) remained relatively stable across different $S_{S,p}$ values, with moderate improvements for $\alpha = 0.7$, 1.0, and 5.0. In contrast, simulations with $\alpha = 0.1$ yielded negative NSE values (Table 2). The highest $\mathrm{NSE}_{\delta^2\mathrm{H}}$ values—approximately 0.55—were achieved with $\alpha = 1.0$ and $\alpha = 5.0$ for the HOAL catchment. The results in the Wüstebach catchment exhibited a wider range of NSE values, from –11.25 to 0.22, as $S_{S,p}$ increased, suggesting that model performance was more sensitive to the size of the passive storage volume than to the shape factor $\alpha$.





In both catchments, increased passive storage volumes influenced the old tail of transit times ($100 < T < 1000$ days). Increasing $S_{S,p}$ increased the probability of older water contributing to streamflow (Figs. 10, 11) and reduced the fraction of

streamflow younger than 1000 days substantially. The range of differences in the fraction of streamflow younger than 1000 days varied across different mixing assumptions, yet remained consistent overall. In the HOAL catchment (Fig. 10 a-c), under the uniform sampling assumption, the fraction of streamflow younger than 1000 days decreased from 50 % to 5 % as $S_{S,p}$ increased from 500 mm to 5000 mm. Given that model performance remained similar across these scenarios (Table 2), this implies a variability of approximately 45 % in TTD estimation attributable to passive storage volume alone. In the Wüstebach

catchment (Fig. 11 a-c), the corresponding fraction declined from 80 % to 45 %. None of the simulations with $S_{S,p}$ less than 5000 mm adequately reproduced the observed $\delta^2$H signal, suggesting that at least 50 % of stream water in Wüstebach is older than 1000 days.

**Table 2.** Performance metrics for simulated $\delta^2$H values in the HOAL (from 2015 to 2019) and Wüstebach (from 2011 to 2013) catchments under varying passive groundwater storage volumes ($S_{s,p}$) and groundwater SAS function shape parameters ($\alpha$). For each $S_{s,p}$ volume (500 mm, 1000 mm, and 5000 mm), simulations were run with $\alpha$ values representing a range from very young-water preference ($\alpha = 0.1$) to old-water preference ($\alpha = 5.0$). The root zone SAS function was fixed at its calibrated value for each catchment ($\alpha_0 = 0.14$ for HOAL and 0.98 for Wüstebach). Performance was evaluated using the Nash–Sutcliffe Efficiency ($NSE_{\delta^2H}$) and Mean Absolute Error ($MAE_{\delta^2H}$) between observed and simulated streamflow $\delta^2$H signals.

| Catchment | Metric | $S_{s,p} = 500\,mm$ | | | | $S_{s,p} = 1000\,mm$ | | | | $S_{s,p} = 5000\,mm$ | | | |
|---|---|---|---|---|---|---|---|---|---|---|---|---|---|
| | $\alpha =$ | 0.1 | 0.7 | 1.0 | 5.0 | 0.1 | 0.7 | 1.0 | 5.0 | 0.1 | 0.7 | 1.0 | 5.0 |
| **HOAL** | $NSE_{\delta^2H}$ | -1.20 | 0.49 | 0.55 | 0.55 | -1.02 | 0.53 | 0.55 | 0.56 | -0.68 | 0.55 | 0.56 | 0.56 |
| | $MAE_{\delta^2H}$ | 5.34 | 2.92 | 2.60 | 2.67 | 5.00 | 2.70 | 2.51 | 2.50 | 4.55 | 2.53 | 2.49 | 2.49 |
| **Wüstebach** | $NSE_{\delta^2H}$ | -11.25 | -12.52 | -13.34 | -12.17 | -5.77 | -7.58 | -6.88 | -8.08 | 0.14 | -0.44 | 0.31 | 0.22 |
| | $MAE_{\delta^2H}$ | 3.70 | 3.74 | 4.04 | 3.64 | 2.77 | 3.05 | 3.04 | 2.84 | 0.77 | 1.16 | 0.76 | 0.81 |



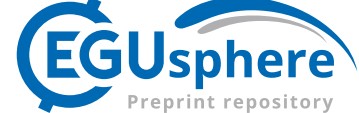

**Figure 10.** Simulated $\delta^2$H signals in streamflow (Q; mm d$^{-1}$) for the HOAL catchment in the year 2015, based on varying passive groundwater storage volumes ($S_{S,p}$ = 500 mm, 1000 mm, and 5000 mm) and different mixing assumptions defined by SAS function shape parameters ($\alpha$ = 0.1, 0.7, 1.0, and 5.0). (a-c) each plot shows results for one $S_{S,p}$ value, with black dots indicating observed grab samples of streamflow $\delta^2$H, and coloured lines representing simulated $\delta^2$H under the different $\alpha$ values. The inset in each plot shows the mean empirical cumulative distribution functions (CDF) of simulated daily streamflow transit times during the tracking period (2015–2019); line colours correspond to $\alpha$ values: blue for 0.1, turquoise for 0.7, purple for 1.0, and red for 5.0. Simulations over the full tracking period (2015–2019) are provided in the Supplement S 9.





**Figure 11.** Simulated $\delta^2$H signals in streamflow (Q; mm d$^{-1}$) for the Wüstebach catchment in the year 2011, based on varying passive groundwater storage volumes ($S_{S,p}$ = 500 mm, 1000 mm, and 5000 mm) and different mixing assumptions defined by SAS function shape parameters ($\alpha$ = 0.1, 0.7, 1.0, and 5.0). (a-c) each plot shows results for one $S_{S,p}$ value, with black dots indicating observed grab samples of streamflow $\delta^2$H, and coloured lines representing simulated $\delta^2$H under the different $\alpha$ values. The inset in each plot shows the mean empirical cumulative distribution functions (eCDFs) of simulated daily streamflow transit times during the tracking period. Line colours correspond to $\alpha$ values: blue for 0.1, turquoise for 0.7, purple for 1.0, and red for 5.0. Simulations over the full tracking period (2011–2013) are provided in the Supplement S 10.





## 4 Discussion

### 4.1 Comparison of catchment transit times

The inferred transit times in HOAL (13 % of streamwater younger than 1000 days) and Wüstebach (27 % of streamwater younger than 1000 days) indicated that, in both catchments, the majority of water contributing to streamflow was relatively old—consistent with findings from many other catchments (Kirchner et al., 2023; Floriancic et al., 2024; Wang et al., 2025). During wet periods, the fraction of water $T < 1000$ days was 15 % in HOAL and 33 % in Wüstebach; in dry periods, these values dropped to 10 % and 22 %, respectively. This variation indicated a greater release of younger water under wetter conditions,

consistent with other studies (Klaus et al., 2013; Angermann et al., 2017; Loritz et al., 2017). In Wüstebach, relatively high soil moisture and high monthly mean young-water fractions ranging from 5 % to 15 % (Figs. 4e, f) pointed to wet-soil promotion of preferential flow which has been observed previously (Wiekenkamp et al., 2016; Stockinger et al., 2014; Hrachowitz et al., 2021; Hövel et al., 2024). By contrast, HOAL's younger-water release did not depend on soil moisture only; instead, rapid flow pathways (e.g. infiltration-excess overland flow, macropores, tile drains) as known for this catchment (Exner-Kittridge

et al., 2016; Pavlin et al., 2021; Vreugdenhil et al., 2022) allowed water to bypass much of the soil matrix and reach the stream quickly, even under dry conditions, which is consistent with previous findings (Türk et al., 2024; Széles et al., 2020). Given the consistency of our results with prior tracer-based modelling and SAS applications in both HOAL (Széles et al., 2020; Türk et al., 2024) and Wüstebach (Stockinger et al., 2019; Hrachowitz et al., 2021), the applied model configurations were considered reasonable and reliable for testing the research hypotheses in this study.

### 355 4.2 Do stream water tracer data have sufficient information content to identify preferential flow in the unsaturated root zone and in groundwater using different SAS functions?

The stepwise analysis presented in Figure 5 and Table 1 indicated that streamflow tracer data were sufficiently sensitive to identify preferential flow in the shallow unsaturated zone for the HOAL and the Wüstebach catchments. Specifically, changing the root-zone SAS shape parameter $\alpha_0$ produced clear differences in simulated streamflow $\delta^2$H signals (Fig. 5), demonstrating

tracer data sensitivity to young water release via shallow subsurface pathways (Stockinger et al., 2016; Benettin et al., 2017). Positive correlations and model efficiency metrics at lower $\alpha_0$ values (indicating a preference for younger water, Table 1) supported this interpretation, indicating rapid transport of precipitation through preferential routes in both catchments, which is consistent with previous findings (Wiekenkamp et al., 2016; Stockinger et al., 2014; Széles et al., 2020).

Nevertheless, there were differences in the processes controlling preferential flow. The calibrated lower boundary of the SAS

function shape parameter, $\alpha_0$, differed between the two catchments ($\alpha_0 = 0.14$ for HOAL and $\alpha_0 = 0.98$ for Wüstebach), suggesting distinct internal catchment characteristics that control the storage–discharge relationship in the unsaturated root zone and preferential flow activation. For HOAL, an $\alpha_0$ of 0.14 indicated rapid, direct overland flow processes driven by intense rainfall, consistent with previous hydrometric analyses and field observations (Pavlin et al., 2021; Vreugdenhil et al., 2022). This rapid flow was further facilitated by soil crust formation and cracking of the clay-rich topsoil, creating direct

preferential pathways that quickly transport water through the catchment (Exner-Kittridge et al., 2016). In contrast, the forest



cover in Wüstebach promotes higher infiltration rates, enhancing subsurface mixing (Wiekenkamp et al., 2016) compared to the HOAL catchment. Despite these differences, findings from both catchments align with previous studies that have generally documented the importance of macropores and preferential flow pathways in the unsaturated zone, highlighting that water frequently bypasses matrix storage and exchange processes (Zehe et al., 2006; Angermann et al., 2017; Sprenger et al., 2016;

Klaus et al., 2013; Loritz et al., 2017), as it was reflected in the SAS formulation through model calibration (Hrachowitz et al., 2013; van der Velde et al., 2015). These results underline the need to calibrate, rather than assume, root zone SAS parameters when estimating transit times.

The Spearman rank correlations ($r$) between observed and simulated $\delta^2$H were lower in HOAL compared to Wüstebach, which can be attributed in part to differences in temporal resolution and the variability of isotope sampling. In HOAL, streamflow

$\delta^2$H was sampled on an event basis, with values ranging from $-26.2$ ‰ to $-108.0$ ‰ (Fig. 1b). In contrast, the Wüstebach catchment weekly to biweekly sampling scheme, yielding streamflow $\delta^2$H values between $-45.6$ ‰ and $-57.1$ ‰ (Fig. 1b). Although model performance metrics such as NSE or correlation coefficients quantify the agreement between simulated and observed isotope time series, they can result in seemingly good fits in the presence of sparse or irregular data sampling (Beven, 2006) and influence TT estimation (Stockinger et al., 2016)

Streamflow tracer $\delta^2$H showed limited sensitivity to variations in groundwater SAS function shape parameters (Fig. 7, Table 1). Our conceptual tracer-based model simulations indicated minimal differences in simulated $\delta^2$H signals across various groundwater SAS shapes, including accounting for preferential flow in groundwater. In the HOAL catchment, the Spearman rank correlation ($r$) varied slightly (0.54–0.60) among different groundwater $\alpha$ values. Similarly, correlations remained relatively stable in the Wüstebach catchment, peaking at $r = 0.76$ for $\alpha = 1.0$, but showing little variation overall. These results suggest

that $\delta^2$H tracer data alone may not carry sufficient information to clearly distinguish preferential groundwater flow dynamics at the catchment scale.

We attributed this insensitivity to the substantial passive groundwater storage volumes (3117 mm in HOAL and 9976 mm in Wüstebach). In our and many other catchment scale modelling approaches (Benettin et al., 2015a; Hrachowitz et al., 2013; Wang et al., 2023, 2025), groundwater ($Q_S$) age selection is formulated based on age samples from the total groundwater

storage ($S_{S,\text{tot}}$), combining contributions from both active ($S_{S,a}$) and passive ($S_{S,p}$) compartments (Zuber, 1986; Hrachowitz et al., 2015). Thus, the age-ranked groundwater storage ($S_{T,S,\text{tot}}$) inherently reflected a mixture of these storage volumes. Substantial passive storage characterised by long residence times strongly buffers the isotopic signals in streamflow (Birkel et al., 2011a), effectively masking distinct signatures of preferential groundwater flow, as is the case in our catchments too. Although our model explicitly allowed preferential recharge of younger groundwater(e.g, $R_{fs}$, Fig. S 1), subsequent mixing

within the large passive groundwater storage dampened $\delta^2$H in streamflow. Consequently, varying the groundwater SAS shape had negligible effects on simulated streamflow $\delta^2$H dynamics within the parameter ranges tested here.

The sensitivity of tracer signals to passive storage volumes further underscored the uncertainty introduced by the conceptual storage parameters. In both catchments, the isotope signals were substantially dampened when the volume rate between active and passive storages ($S_{S,a}/S_{S,p}$) fell below 1 %. Even maximal mixing between compartments thus appeared sufficient to

markedly reduce isotopic variations, particularly when large passive volumes buffered hydrological responses. Nevertheless,





the absolute storage volume required to achieve the observed isotope damping differed notably between the two catchments. In HOAL, model performance remained stable across a wide range of passive storage volumes above 500 mm (e.g., $\text{NSE}_{\delta^2 H} \approx$ 0.55 for both $S_{S,p} = 500$ mm and $S_{S,p} = 6117$ mm), suggesting high uncertainty in estimating the upper bound of passive storage. In contrast, model performance in Wüstebach improved significantly with larger passive storage volumes (Table 2),

consistent with previous observations by Hrachowitz et al. (2021), who argued for substantial passive storage ($\sim 8000$ mm) to replicate observed isotope damping patterns.

An alternative explanation, however, must also be considered: it is possible that such preferential groundwater flow processes are simply absent or negligible in the HOAL and Wüstebach catchments. The current data and model structure are insufficient to conclusively rule out either possibility. Ultimately, distinguishing between limitations in model sensitivity and the actual

absence of preferential flow processes requires additional, spatially distributed tracer data and complementary hydrometric observations.

## 4.3 Does accounting for preferential groundwater flow (and associated SAS functions) affect catchment-scale transit time distributions?

SAS function shape changes in groundwater only marginally affected model performance (Table 2), with the exception of the

strong young-water preference ($\alpha = 0.1$) in the HOAL catchment. This suggests that differences in mixing assumptions had limited influence on model fit. However, the associated transit time distributions (TTDs) were substantially different, and thus, the estimation of TT is uncertain. Consistent with previous findings (van der Velde et al., 2012; Borriero et al., 2023), our results highlighted that TTD estimates are highly sensitive to how SAS functions are conceptualised and parameterised within the model.

Specifically, the empirical cumulative distribution functions (eCDFs) of simulated TTDs (Fig. 9) revealed notably different ranges: assuming a strong young-water preference ($\alpha = 0.1$) resulted in the fraction of streamflow younger than 1000 days to approximately 25 % in HOAL and 35 % in Wüstebach, whereas an older-water preference ($\alpha = 5.0$) reduced these fractions to around 5 % and 12 %, respectively. This variability in transit time estimations for T< 1000 days —roughly 20 % for HOAL and 23 % for Wüstebach—underscores a critical limitation in modelling groundwater transit times.

Given these uncertainties arising from groundwater SAS function shapes alone, it is also crucial to assess how passive groundwater storage volumes, through their mixing with active groundwater storage, further modulate transit time estimates. Our results indicated that passive storage volumes substantially influenced the longer tails of the inferred TTDs, highlighting their importance in catchment transit time estimation. In both catchments, we observed a clear negative correlation between passive storage volume and the fraction of streamflow younger than 1000 days (Figs. 10, 11). For instance, under a uniform sampling

assumption, the fraction of young water decreased markedly—from approximately 45 % to 10 % in HOAL and from about 85 % to 25 % in Wüstebach—as passive storage increased from 500 mm to 5000 mm (Fig. 11). Since the SAS function was formulated based on age-ranked total groundwater storage ($S_{S,\text{tot}} = S_{S,a} + S_{S,p}$), larger passive storage volumes increased the probability of older water contributions to streamflow.





The methodological framework applied here, including the stepwise analysis of SAS functions and the incorporation of multiple
passive storage volumes, offered a systematic approach that could be adapted to other regions and TTD studies. Nonetheless,
the uncertainty resulting from the specific model setup and parameter choices used in this study cannot be directly generalised
across diverse catchments or hydrological conditions. Addressing these limitations, e.g., by improving monitoring frequency of
(isotope) hydrological data, integrating additional tracers such as tritium ($^3H$), and refining model representations of subsurface
processes, will be essential for reducing uncertainty and enhancing the reliability of SAS-based modelling.

**5  Conclusion**

In this study, we evaluated whether stream water isotope data provide adequate information to identify and quantify preferential
flow in the unsaturated zone and in groundwater using various StorAge Selection (SAS) function shapes at the catchment
scale. We further analysed the implications of explicitly representing preferential root zone and groundwater flow and passive
storage volumes on the estimation of transit time distributions (TTDs). Our findings underscore critical limitations in using
isotope tracer data alone to constrain groundwater transit times, emphasising the influence of passive groundwater storage on
uncertainty in catchment-scale models. The main findings of our study are:

- Streamflow isotope ($\delta^2$H) data were sensitive enough to characterise preferential flow processes in the unsaturated root
  zone, confirming that such processes significantly shape catchment isotope signatures and transit time distributions at
  short timescales (up to 300 days). This highlighted the need to calibrate, rather than assume, root zone SAS parameters
when estimating transit times.

- Streamflow isotope data alone were insufficient to differentiate among groundwater SAS function shapes for the two
  tested catchments. Large passive groundwater storage volumes significantly dampened isotopic variations, making it
  impossible to clearly identify preferential flow in groundwater.

- The variability in groundwater TTD estimates arising from varying SAS function shapes for groundwater was consid-
erable (20 % for HOAL and 23 % for Wustebach), highlighting that TTD estimates are highly sensitive to how SAS
  functions are conceptualised and parameterised within the model.

- Passive groundwater storage volumes strongly controlled the catchment transit time distributions, particularly affecting
  the longer tails ($T > 100$ days). Increasing passive storage reduced the fraction of younger streamflow ($T < 1000$ days),
  introducing uncertainties into solute and contaminant transport predictions.

These findings carry implications beyond water transit times, also affecting the transport timescales of solutes and contam-
inants within catchments. Larger passive storage volumes imply prolonged retention times, potentially delaying pollutant
transport and release. Consequently, uncertainty in estimating passive storage volumes directly translates into uncertainty
regarding contaminant transport predictions, with critical implications for assessing water-quality risks. Additional or com-
plementary datasets—such as direct groundwater measurements or higher-frequency tracer sampling—would be required to

reliably characterise preferential groundwater flow using conceptual catchment-scale models. Improved characterisation of passive storage volumes—potentially via complementary observations (e.g., groundwater-level monitoring or high-frequency isotope sampling)—is essential to reduce uncertainties and enhance reliability in transit time and solute transport modelling at the catchment scale.

*Code and data availability.* A Python script that performs the calculations described in this paper will be deposited in an open-access Github
archive repository, and the link will be supplied with the final published paper. The code repository for the *Tracer Transport Model* is available on GitHub at: https://github.com/haticeturk/Tracer_Transport_Model.git. Model outputs, including state variables, fluxes, hydrological signatures, parameter sets, and performance metrics underlying this study, are available online in the FAIR-compliant Zenodo repository. The meteorological and hydrological data from the Wüstebach TERENO site used in this study are openly accessible through the *Terrestrial Environmental Observatories (TERENO)* of the *Helmholtz Association of German Research Centers (HGF), Germany*, via the TEODOOR
data portal (http://teodoor.icg.kfa-juelich.de/). The stable water isotope dataset for the Wüstebach catchment is publicly available through a digital object identifier (DOI) at: https://doi.org/10.34731/y6tj-3t38 (Bogena et al., 2021). The data for the HOAL catchment can be available from the Austrian Federal Agency for Water Management upon request.

*Author contributions.* HT performed the analysis presented here and drafted the paper. All authors discussed the design, contributed to the overall concept, and participated in the discussion and writing of the manuscript.

*Competing interests.* Some authors are members of the editorial board of the HESS journal.

*Acknowledgements.* We acknowledge the *Terrestrial Environmental Observatories (TERENO)* of the *Helmholtz Association of German Research Center (HGF), Germany*, for providing access to the Wüstebach catchment data. We thank the Austrian Federal Agency for Water Management for providing the data the HOAL catchment that we used in our analysis. This research was funded by the *Austrian Science fully Fund by (FWF – Österreichischer Wissenschaftsfonds)* [Grant No. 10.55776/P34666]. For open access purposes, the author has applied
a CC BY public copyright license to any author accepted manuscript version arising from this submission. The work of Hatice Turk was supported by the Doctoral School "Human River Systems in the 21st Century (HR21)" of the BOKU University, Vienna.



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
