# Peer review of "Catchment transit time sensitivity to the type of SAS function for unsaturated zone and groundwater"

_EGUsphere, 2025_

## Author Comment (AC1)

**Reviewer #1**

**Comment:** Thank you for the opportunity to review the manuscript entitled "Catchment transit time sensitivity to the type of SAS function for unsaturated zone and groundwater." The authors present an interesting study comparing model performance across different setups that vary in the parameters used in the SAS function. This approach aims to deepen the understanding of catchment functioning, specifically the contributions of water from the unsaturated zone and groundwater to catchment transit times in two catchments. Overall, the manuscript addresses an important research gap by challenging an assumption in transit time distribution (TTD) modeling and has the potential to make a valuable contribution. However, I believe that the manuscript in its current form requires major revisions before it can be considered for publication. My main concerns are as follows:

**Response:** We thank Reviewer #1 for the evaluation and positive feedback on our manuscript. Below, we outline how we consider the issues raised by the reviewer and the changes we intend to make. Reviewer comments are shown in black, while our responses are shown in blue.

**Comment:** The language used throughout many sections of the manuscript is often vague and imprecise, which unfortunately leads to several hydrologically inaccurate statements. For example, there is some conflation between residence time distribution and transit time distribution (line 290), the SAS function is described as specifying if rather than how young or old water leaves the storage (line 6), and terms commonly well-defined in hydrology, such as information content (line 10) and "process", are used incorrectly. A thorough revision of terminology is necessary, particularly in the abstract and introduction, to clarify these points and improve the manuscript's overall precision.

**Response:** We thank the reviewer for this comment. We will revise the manuscript to improve precision and consistency in terminology and ensure that information content and process are used in accordance with established hydrological terminology throughout the entire manuscript.

For example Line 6 (Abstract): The original text "functions that specify if young or old water leaves a storage" will be revised to:"…functions that specify how young or old water leaves a storage";

Line 290 (Results): The original text "…influenced the TTD for ages up to 300 day. This was due to the fact that root-zone storage residence time remained predominantly younger than 300 days" will be revised to: "…influenced the transit time distribution (TTD) for ages up to 300 days. This is consistent with the age of the water in root zone storage (i.e., the residence time distribution) being predominately younger than 300 days, and as TTD are part of resident water leaving the system."

In addition, we will add a brief definition of residence time at its first mention in the manuscript to avoid confusion with transit time: "residence time refers to the duration that water remains in a given storage before leaving it.

**Comment:** The hypotheses as currently formulated are not effectively integrated into the manuscript. First, the connection between the hypotheses and the discussion is missing and could be strengthened by revising the discussion section, which currently has some weaknesses (see one of my later comments). Second, the hypotheses are not formulated in a testable manner, which limits their usefulness in framing the study.

**Response:** We thank the reviewer for this comment. We will revise the formulation of the hypotheses to make them more precise and testable. The revised text will read: "In this study, we hypothesize that preferential release of young water in unsaturated zone and in groundwater contributes measurably to streamflow tracer signal. This effect can be represented through SAS functions, i.e., functions that specify how young or old water leaves a storage, parameterized to favor the release of younger water from storage in catchment-scale transport models". We further adapted the discussion to address the hypotheses more explicitly.

**Comment:** Although the results section is generally well-written, the key messages are often difficult to discern. A clearer focus and synthesis of the main findings would greatly enhance the manuscript's impact.

**Response:** We thank the reviewer for this comment. We will revise the Results section to more highlight the key messages by adding short synthesis statements at the end of relevant subsections, ensuring that the main findings are explicitly summarized.

**Comment:** I am uncertain about the comparability of the different model setups as currently presented. The authors rely on common performance metrics such as NSE for comparison, but their stated hypotheses suggest an information theory or information content approach. If the authors choose to maintain the current comparison method, they should provide a more detailed explanation and justification for this choice. NSE also has certain limitations, which should be taken into account in this context.

**Response:** We thank the reviewer for this comment. We will revise the hypotheses to focus on the variability in tracer and streamflow data rather than on an information theory framework. Consistent with this, tracer transport models are generally evaluated based on tracer fit. We therefore assessed model performance using NSE, MAE, and correlation coefficients (r values), which together capture both variability and error characteristics across scenarios, while each metric also has its limitations. We agree that NSE alone can be misleading, and we will clarify this in the methods while justifying our choice of combining NSE, MAE, and r as a balanced evaluation approach. In addition, we will highlight in the discussion that weekly-resolution δ²H measurements can lead to deceptively high NSE values, even when key groundwater age-selection parameters (e.g., preference for young vs. old water) remain poorly constrained. We will add the method section 2.3.1 Model calibration and evaluation:

"Model performance was evaluated using Nash–Sutcliffe Efficiency (NSE), Mean Absolute Error (MAE), and correlation coefficients (r). These metrics were selected because they are commonly used in tracer transport modeling and capture different aspects of model fit: NSE emphasizes explained variance, MAE quantifies absolute deviations, and r assesses correlation structure. While each metric has limitations when considered alone, their combined use provides a more balanced basis for comparing model setups."

**Comment:** The discussion would benefit from a broader perspective. Addressing questions such as the following could improve the manuscript's relevance and clarity: What are the key findings for the research community? What are the broader implications of this work? How does this study advance process understanding in other, unstudied catchments? Under what conditions might one expect similar or different results?

**Response:** We thank the reviewer for this suggestion. We will add a subsection on Limitations and Implications in the Discussion to provide a broader perspective. Part of it will be the following text:

"These findings emphasize both the opportunities and limitations of tracer-informed SAS modeling. While our study focused on a lumped catchment-scale framework, the results highlighted the need for advancing toward more distributed models that can more directly link spatial heterogeneity in soils, slopes, and storages to preferential flow dynamics, and for extending such approaches. Catchment scale, isotope-based modeling proved useful in capturing preferential flow in the unsaturated zone but was limited in groundwater due to damping of the seasonal signal of the water stable isotopes by large passive storage volumes assumed through the models. This suggests that in catchments with similarly damped water stable isotope signals, groundwater age selection will be difficult to constrain, whereas in systems with smaller storage, additional tracers or higher-frequency sampling could help to better distinguish groundwater mixing processes and age selection."

**Comment:** While the authors refer to a previous study for a description of the model setups, I believe the manuscript would benefit from a brief summary of the underlying assumptions and structural details within the current text. This would enhance clarity for readers unfamiliar with the referenced work.

**Response:** We will add a brief summary of the underlying assumptions and structural details of the model to enhance clarity for readers unfamiliar with the referenced work. In addition, we will include the relevant bucket-model equations for the water age balance (currently presented in Türk et al., 2024, Section "Integration of the rSAS function concept and the hydrological model") and provide a clearer description of how each bucket's tracer balance is formulated and parameterized through the SAS framework.

**Comment:** Additionally, the configuration used to test passive groundwater storage requires further explanation. In particular, it would be helpful if the authors could elaborate on the rationale for choosing the specific storage volumes and clarify how mixing within this storage compartment was considered.

**Response:** We thank the reviewer for this comment. We will expand the description of the passive groundwater storage setup to clarify both the rationale for the selected volumes and how mixing was implemented. Specifically, we will explain that groundwater was represented as consisting of an active ($S_{s,a}$) and a passive ($S_{s,p}$) storage, where $S_{s,p}$ mixes isotopically with active storage but does not contribute directly to runoff. The total storage ($S_{s,tot}$) therefore reflects both compartments, influencing the tracer signal of baseflow. We will also clarify that the three passive storage volumes ($S_{s,p}$ = 500, 1000, and 5000 mm) were chosen to cover the range reported in comparable headwater catchments (Birkel et al., 2011; Benettin et al., 2015; Hrachowitz et al., 2021; Wang et al., 2023).

Some specific comments are provided below for consideration.
* * *
**Comment:** Title "sensitivity": It is not clear whether the manuscript presents a sensitivity analysis. Please clarify if such an analysis was performed; otherwise, consider revising the title accordingly.

Title "type of SAS function": Typically, SAS function types refer to, for example, gamma or beta functions. This may not be what you mean here please clarify the intended meaning.

**Response:** We will revise the title to reflect the scope of the study. Since our work compares different SAS parameterizations rather than performing a formal sensitivity analysis, we will replace "sensitivity" with wording that avoids potential misinterpretation. We will also clarify that by "type of SAS function" we refer to parameterizations favoring younger versus older water leaving storage, not to specific functional forms.

**Revised Title:**
"Catchment transit time variability with different SAS function parameterizations for the unsaturated zone and groundwater"

Line 3: The terms fast and short are subjective and relative. Please avoid judgmental terms.

**Response:** We will revise the sentence as to avoid judgmental terms

"facilitate more rapid water and solute transport, leading to quick streamflow responses and comparatively short water transit times.

Line 3: In the phrase "such preferential flow processes," please clarify which specific processes from the previous statements are meant—do you mean preferential flow paths?

**Responce:** We will revise the sentence and replace "such preferential flow processes," to" preferential flow paths" as:

"While preferential flow paths are well documented in the unsaturated zone and groundwater…"

Line 4: Please clarify the reference of the word "these."

**Responce:** We will remove "these" from the sentence.

Line 5: Use the term significant only where statistical significance (p-value) is reported; otherwise, rephrase.

**Responce:** We will remove the term "significant" and rephrase the hypothesis to make it more testable and precise.

Line 5: The statement "…by selecting specific SAS functions" seems self-evident because preferential discharge of young groundwater cannot be represented without selecting an SAS function. Please refine the hypothesis so that it is testable and cannot be answered with a simple "yes."

We agree with this comment, and we will revise the hypothesis as: "In this study, we hypothesize that preferential release of young water in unsaturated zone and in groundwater contributes measurably to streamflow tracer signal. This effect can be represented through SAS functions, i.e., functions that specify how young or old water leaves a storage, parameterized to favor the release of younger water from storage in catchment-scale transport models"

Line 6: Replace "if" with "how."

**Responce:** We will replace "if" with "how."

Lines 6–8: It is unclear whether the functions were parameterized or if they were part of different model setups. Please clarify.

**Responce:** We agree that the original wording was unclear. We will revise the sentence to explicitly indicate that it was the SAS functions that were parameterized, rather than different model setups. To clarify this, we will revise the sentence to read as follows:

"We systematically compared multiple parameterizations of the StorAge Selection (SAS) functions for the unsaturated zone and groundwater within a catchment-scale transport model using long-term measurements of hydrogen isotopes ($\delta^2H$) from two headwater catchments (the Hydrological Open Air Laboratory (HOAL) in Austria and the Wüstebach catchment in Germany)."

Line 10: The term "information content" implies the use of information theory; please clarify if that is the case.

**Responce:** Thank you for the comment. Our intention was to express that the $\delta^2H$ ratios in streamflow exhibited sufficient variability/sensitivity to preferential flow in the unsaturated zone. To clarify this, we will revise the sentence as follows:

"The results indicated that $\delta^2H$ ratios in streamflow exhibited significant variability to identify preferential flow in the unsaturated zone. This was supported by Spearman rank correlations (r) between simulated and observed $\delta^2H$ signals in streamflow, where r values ranged between 0.58 and −0.18 for HOAL, and between 0.58 and 0.28 for the Wüstebach catchment, reflecting a transition from strong young-water preference to old-water preference across SAS shape parameter values between 0.1 and 5.0."

Lines 10–18: This section should be improved by clearly stating the main message of the results and explicitly explaining the relationship between age and specific SAS functions.

**Response:** We will revise lines 10–18 to more clearly state the main message of the results and explicitly link the findings to the role of SAS functions. The revised text will read:

"However, $\delta^2H$ ratios in streamflow showed limited variability to identify preferential release of young water in groundwater. In the HOAL catchment, r values ranged only between 0.54 and 0.60, while in the Wüstebach catchment they varied between 0.71 and 0.76 across SAS shape parameter values from 0.1 to 5.0. This limited sensitivity reflected that seasonal variation in $\delta^2H$ in pore water was strongly dampened by the catchments' substantial passive groundwater storage volumes. This was further confirmed as the observed attenuated $\delta 2H$ signal in streamflow could only be simulated when the volume ratio between active and passive groundwater storage was < 1 %, highlighting the dependence of SAS-based age selection on storage configuration. This damping effect, combined with the groundwater SAS function parameterization, influenced the estimation of the longer tails ($100 < T < 1000$ days) of the transit time distributions, making it difficult to determine the fraction of stream water older than 100 days."

Line 24: The term "dry period" depends on climatic context; please specify.

**Response:** We agree that the term "dry period" is too general and depends on climatic context. We will revise the sentence to clarify that we refer to low-flow conditions resulting from below-average precipitation and/or reduced soil moisture availability. The revised sentence will read:

"Groundwater plays a crucial role in the hydrological cycle and in sustaining streamflow during low-flow periods, thus regulating the timing and quality of baseflow contributions to streams (van der Velde et al., 2011; Hamilton, 2012; Kaandorp et al., 2018b)."

Line 25: Specify whether "reaching streams" refers to water after precipitation events or baseflow contributions.

**Response:** We will revise the text by replacing "quality of water reaching streams" with "quality of baseflow contributions to streams."

Lines 28–30: As written, "this variability" (temporal) cannot be caused by spatial factors such as catchment topology. You may mean differences in temporal variability among catchments. Please reframe.

**Response:** We will replace "this variability" with "variation in flow timescales across catchments" for clarity.

Line 33: The word "dramatically" is subjective; please remove.

**Response:** We will remove the word "dramatically" from the sentence.

Line 38: A model, by definition, cannot detect anything: consider alternative terminology. Similarly, "quantify" may not be applicable in this context.

**Response:** We agree that "detect" and "quantify" are not the correct terminology here. We will revise the sentence to: "transport models can meaningfully simulate preferential groundwater flow pathways.

Lines 40–41: The terms "follow" and "system" are too vague: please specify.

**Response:** We will revise the sentence for clarity. The revised version will read: "Water molecules entering at different locations within a catchment travel along distinct flow paths and exit at different times via streamflow or evaporation (transit time, TT).

Line 43: Consider using "control volumes, such as catchments" instead of just "a catchment," since the approach could also be applied to a lysimeter or a stream reach.

**Response:** We will revise the text to read "control volumes, such as catchments

Line 46: A process cannot be quantified directly from a TTD; the TTD allows you to infer processes. Please check other occurrences where "process" is used in a similar way.

**Response:** We agree that a process cannot be quantified directly from a TTD. We will revise the sentence to clarify:

"Many studies have integrated hydrometeorological data and applied tracer-based modelling, using the transit time distribution (TTD) to infer flow processes and estimate transit times."

Line 48: Please clarify why "most" applies here.

**Response:** We will remove the word "most."

Line 49: The message of this sentence is not sufficiently clear. Please rephrase.

**Response:** We will rephrase the sentence for clarity as follows:

"These studies have shown that streamflow typically consists of water from a broad spectrum of ages, with transit time distributions (TTDs) spanning from days to decades, thereby highlighting the importance of both rapid flow pathways and long-term storage in catchments."

Line 53: Keep terminology consistent. Use either function or model only when referring to different concepts. Here, it should be "SAS function." Also, note that the SAS function does not directly capture storage heterogeneity; please define precisely.

**Response:** We will revise the sentence to: "The SAS function represents water age dynamics in hydrological systems by defining the relationship between the distribution of ages stored (residence time distribution, RTD) and the ages removed as outflows (transit time distribution, TTD)"

Line 55: Same adjustment as above. Use function instead of model if referring to the SAS function.

**Response:** We will replace "model" with "function" when referring to the SAS function.

Line 57: Consider rephrasing, as it is unsurprising that time-variable TTDs reflect temporal TTD variability.

**Response:** We will revise the sentence to: "Studies have shown that TTDs vary over time and that transport processes can differ under contrasting conditions, such as between wet and dry periods (Benettin et al., 2015b; Harman, 2015; Kaandorp et al., 2018a)."

Lines 59–63: Please re-check whether all cited studies actually applied SAS functions.

**Response:** We will re-check the cited studies and ensure that only those that applied SAS functions are referenced in this context.

Lines 66–67: Please explain more clearly what is meant by "the age composition of groundwater flow to the stream."

**Response:** We will revise the sentence to: "In many SAS function applications, the age distribution of baseflow (groundwater contribution to streamflow) is simplified by assuming uniform mixing of stored ages."

Line 69: SAS functions cannot be "measured"; you can parameterize them. Please revise.

**Response:** We will revise the sentence to: "Noting that SAS functions are not straightforward to parameterize."

Line 79: Specify what the "release of young water" refers to.

**Response:** We will revise the sentence to: "Indeed, increasing evidence suggests that groundwater systems may not be completely mixed, and that the preferential release of relatively young water (recently recharged water) to streams may be a ubiquitous feature of groundwater in heterogeneous aquifers."

Line 80: Indicate where the "generally low longitudinal and transversal dispersivities" apply.

**Response:** We will revise the sentence to: "… (ii) generally low longitudinal and transversal dispersivities in groundwater systems, leading to little mixing,"

Lines 81–83: This sentence appears disconnected from the previous one; please rephrase for cohesion.

**Response:** We will rephrase the sentence for improved cohesion with the preceding text:

"This evidence suggests that groundwater systems are often not completely mixed and that the preferential release of young water may be common in heterogeneous aquifers. Therefore, SAS functions should be formulated to account for these flow processes and the resulting nonlinearities in groundwater contributions and catchment responses."

Line 93: Clarify whether "long-term tracer observations" were conducted in streams or in groundwater.

**Response:** We will update the sentence to "long-term tracer observations in streamflow."

Line 96: The main objective could be presented in a way that connects more clearly to the underlying processes of interest rather than focusing solely on technical aspects.

**Response:** We will revise the objective to more clearly emphasize the underlying hydrological processes of interest. The revised text will read:

"The main objective of this study was to assess how the young water from preferential flow in the unsaturated zone and in groundwater influences catchment-scale transit times and tracer dynamics. Therefore, we evaluated how different parameterizations of SAS functions, which describe the release of younger versus older water from storage, affect simulated tracer signals at the stream outlet. By systematically comparing multiple mixing assumptions, we tested the hypothesis if preferential release of young groundwater contributes measurably to streamflow tracer signal and if it should therefore be represented in catchment-scale transport models. Additionally, we examined how the extent and mixing of passive groundwater storage modulate tracer signals and shape the tails of the estimated transit time distributions."

Line 103: If the term "information content" is used, ensure that it is correct. If information theory was not applied, please rephrase.

**Response:** We will replace "information content" with "variability" to avoid misinterpretation.

Line 107: Please define what is meant by "interpretation" in this context.

**Response:** We will clarify the wording by replacing "interpretation" with a more precise term like "the ability to reproduce".

"Does explicitly accounting for the preferential release of young water in groundwater, through a SAS function, affect catchment-scale transit time distributions and simulated tracer signals in streamflow?"

Line 108: Clarify the meaning of "representation of preferential groundwater flow."

**Response:** We will revise the research question to avoid the vague term "representation" and to more clearly link passive storage and mixing assumptions to model outcomes. The revised question will read:

"How do different groundwater mixing assumptions, in combination with varying passive storage volumes, affect the model's ability to reproduce streamflow $\delta^2H$ signals, model performance, and the inferred transit time distributions at the catchment scale?"

Line 122: Please state in which catchment the "predominant soil types" are found.

**Response:** We will clarify this by specifying that the predominant soil types are found in the HOAL catchment.

Line 135: The term "catchment flow" is not defined. Do you mean streamflow?

**Response:** Yes, we mean streamflow, and we will revise the text accordingly.

Line 160: It is important to briefly explain the underlying assumptions and general model setup, even if they are covered in a previous publication. Currently, it is unclear how the SAS functions are integrated into the model.

**Response:** We thank the reviewer for this comment. We will revise the sentence as follows to explain the underlying assumptions:

"To route δ²H fluxes through the model, we integrated the storage-age selection (SAS) approach (Rinaldo et al., 2015; Harman, 2015) into the hydrological model. In this framework, each storage is represented as a ranked distribution of water ages, and outflow is sampled from this distribution according to an age-dependent selection function. This allows the model to reproduce different mixing behaviors by allowing either younger or older water leaving storage, depending on the SAS function parameterization."

We will add the relevant bucket-model equations for the water age balance that are described in Türk et al. (2024), Section "Integration of the rSAS function concept and the hydrological model". In addition, we will provide a clearer description of how each bucket's tracer balance is formulated and parameterized through the SAS framework.

Lines 178–184: The SAS function is generally time-variable. Please clarify the novelty of the approach described. Explain why the precipitation value was set as it was and define Sr,max.

**Response:** We will clarify that the dual dependence of α(t) on both soil moisture and precipitation intensity extends previous SAS applications by providing a more flexible representation of unsaturated zone preferential flows in the HOAL catchment. We will also define Sr,max explicitly as the calibrated maximum root-zone storage capacity, which controls the scaling of relative soil wetness in the α(t) formulation. The precipitation threshold value (P_thresh) will be clarified as a calibration parameter representing the intensity above which infiltration excess preferential flow pathways are activated.

We will revise the section as:

"In the HOAL catchment, previous studies highlighted the non-linearity of preferential flow generation, where both precipitation intensity and soil moisture influence the activation of preferential flow pathways (Türk et al., 2024; Széles et al., 2020). To represent this behavior, we parameterized the SAS function for preferential flow from the unsaturated root zone (Rf , Fig. S 1) with a time-variable shape parameter α(t). Specifically, α(t) varied as a function of relative soil moisture (scaled by the maximum root-zone storage capacity, Sr,max) and precipitation intensity (PI, mm d⁻¹), with a threshold parameter (Ptres) controlling the onset of precipitation-driven preferential flow. This formulation allows the model to capture the observed interplay between soil wetness and rainfall events in activating preferential flow. The dual dependence of α(t) on soil moisture and precipitation intensity extends previous SAS applications by providing a more flexible representation of unsaturated zone preferential flow dynamics in HOAL."

Lines 195–199: Please explain why streamflow, log streamflow, flow duration curve, runoff coefficient, and δ²H were selected as variables.

**Response:** We will revise the text to clarify the rationale for selecting these performance metrics. Specifically, the original sentence will be revised as follows:

"For model parameter optimization, we used the Differential Evolution algorithm (Storn and Price, 1997) and an objective function that combined five performance criteria related to streamflow and δ²H dynamics. The objective function included the Nash-Sutcliffe efficiencies (NSE) of streamflow (to evaluate overall discharge dynamics), logarithmic streamflow (to match low-flow conditions), the flow duration curve (to capture the distribution of flows over time), the runoff coefficient averaged over three months (to ensure water balance consistency), and the NSE of the δ²H signal in streamflow (to constrain tracer dynamics) (Table S2). These individual performance metrics were aggregated into the Euclidean distance DE, with equal weights assigned to streamflow and the δ²H signature, according to:"

Line 199: Avoid subjective terms such as "perfect."

**Response:** We will remove the term "perfect" from the sentence

Line 203: Clarify what is meant by "each combination."

**Response:** We will remove the term "each combination" from the sentence for clarity.

Line 211: The need for stepwise calibration and the exact order of steps should be made explicit.

**Response:** To clarify, we did not apply a stepwise calibration but instead used one calibration of the hydrological parameters. The stepwise analysis was then performed afterwards by varying the SAS function parameters ($\alpha_0$ for the unsaturated root zone and $\alpha$ for the groundwater compartments) while keeping the calibrated hydrological parameters fixed. We will revise the text accordingly to avoid confusion.

Lines 218–221: This information may be better placed earlier in the manuscript for clarity.

**Response:** We will move this information earlier in the manuscript to improve clarity.

Lines 228–237: This section should more clearly describe how different passive storage volumes were implemented in the earlier-described model setup and why these volumes were selected. As it stands, the model configuration is not fully reproducible.

**Response:** We thank the reviewer for this comment. We will revise the section to include the following clarification:

"In the model setup, groundwater storage was formulated as consisting of an 'active' groundwater storage ($S_{s,a}$) and a hydrologically 'passive' storage volume ($S_{s,p}$, mm). The passive storage does not change over time if there are no deep infiltration losses (Zuber, 1986; Hrachowitz et al., 2016; Wang et al., 2023). While $S_{s,p}$ does not contribute directly to runoff, it isotopically mixes with water in the active storage, thereby influencing the tracer signal of the outflow. The total groundwater storage ($S_{s,tot}$) was therefore considered as the sum of $S_{s,a}$ and $S_{s,p}$, so that the age-ranked groundwater storage ($S_{T,S,tot}$) inherently reflected a mixture of these storage volumes to baseflow ($Q_s$, Fig S1) age composition."

We will also justify the choice of passive storage volumes as follows:

"Each of these mixing scenarios was applied to three different passive storage volumes ($S_{s,p}$ = 500 mm, 1000 mm, and 5000 mm), which were chosen to cover a ranges reported in comparable headwater catchments (Birkel et al., 2011; Benettin et al., 2015; Hrachowitz et al., 2021; Wang et al., 2023)."

Lines 240–246: Please summarize the key takeaway for the reader.

**Response:** We will add a summary sentence at the end of the section to highlight the main message:

"$\delta^2 H$ in precipitation showed large variability in both catchments, but this signal was damped in streamflow. In HOAL, event-based streamflow $\delta^2 H$ samples reflected how precipitation inputs were transmitted to the stream, whereas weekly samples alone would have masked this variability. In contrast, only weekly streamflow $\delta^2 H$ measurements were available in Wüstebach, resulting in stable values with small variations. This highlights the importance of event-based sampling for detecting preferential flow signals, which may remain obscured with weekly data alone."

Line 248: Define what is meant by "feasible parameter solutions."

**Response:** We thank the reviewer for this comment. We will revise the sentence to clarify that it meant parameter solutions with acceptable model performance (DE < 1):

"Model calibration resulted in 55 feasible parameter solutions (DE < 1, indicating acceptable model performance; Fig. S2) for HOAL and 190 feasible parameter solutions for Wüstebach."

Lines 266–276: Clarify whether these results relate to the stepwise analysis, and connect them to Figure 4. The calculated fractions shown in Figure 4 should also be introduced earlier.

**Response:** We will clarify that these results are not related to the stepwise analysis but are based on the initial model calibration performed prior to the stepwise experiments. We will also introduce the calculated fractions earlier in the text and clearly link them to Figure 4. In addition, we will revise the Methods section to explicitly describe how these fractions were calculated to ensure clarity and reproducibility.

Line 290: The explanation ("This was due to…") belongs to the Discussion. Also, confirm whether "residence time" here should be "transit time," and note that this is the first mention of the term in the manuscript.

**Response:** We will revise the sentence accordingly by removing the explanation and rephrasing it as:

"This is consistent with root-zone storage residence times being predominantly shorter than 300 days."

In addition, we will add a definition of residence time at its first mention to ensure clarity for the reader.

Line 313: Remove the subjective phrase "and somewhat surprisingly."

**Response:** We will remove the phrase "and somewhat surprisingly" from the sentence.

Line 315: Specify which variability is being reduced—e.g., variability in streamflow?

**Response:** We will revise the sentence to clarify that it refers to variability in the $\delta^2H$ signal in streamflow.

Lines 351–354: Consistency with previous findings does not necessarily justify the model configurations or research hypotheses; identical results can occur in the presence of shared erroneous assumptions. Please refine this reasoning.

**Response:** We will refine the reasoning to emphasize that consistency with previous studies provides support but not proof of correctness. The revised text will read:

"The consistency of our results with prior tracer-based modeling and SAS applications in both HOAL (Széles et al., 2020; Türk et al., 2024) and Wüstebach (Stockinger et al., 2019; Hrachowitz et al., 2021) provides additional confidence in the applied model configurations. However, we acknowledge that such consistency alone cannot exclude the possibility of shared structural assumptions. Therefore, we use these results as supporting evidence that the model setups are reasonable for testing the research hypotheses, while recognizing the need for further validation with complementary data and approaches."

Lines 357–364: Please state clearly what the novelty of this section is.

**Response:** We will revise the text to highlight the novelty of this section by adding the following sentence at the end:

"The novelty of this analysis is that it provides the first systematic demonstration that stable isotope data can constrain SAS parameterizations of root-zone preferential flow in these two contrasting catchments."

Line 369: Clarify whether the described crust formation was observed directly, or if it is inferred.

**Response:** We will revise the sentence to clarify that the crust formation was observed:

"During the dry summer months, cracks often form in the soil.."

Line 373: Make the link to the previous sentence explicit.

**Response:** We will rephrase the sentence to make the connection explicit:

"These contrasting site characteristics led to consistent findings though, as results from both catchments aligned with previous findings documenting the importance of macropores and preferential flow pathways in the unsaturated zone."

Line 376: Clarify who assumes this.

**Response:** We thank the reviewer for this comment. We will revise the sentence to avoid the ambiguous term "assume". The revised text will read:

"These results underline the need to calibrate root-zone SAS parameters when estimating transit times, as the tracer signal showed sensitivity to the SAS function, implying that these parameters can be calibrated."

Lines 419–444: There is considerable repetition of the results in this section. Consider condensing.

**Response:** We agree with this comment and will condense the section accordingly

**References:**

Asadollahi, M., Stumpp, C., Rinaldo, A., and Benettin, P.: Transport and water age dynamics in soils: A comparative study of spatially integrated and spatially explicit models, Water Resour. Res., 56 e2019WR025539, https://doi.org/10.1029/2019WR025539, 2020.

Birkel, C., Soulsby, C., and Tetzlaff, D.: Modelling catchment-scale water storage dynamics: Reconciling dynamic storage with tracer inferred passive storage, Hydrological Processes, 25, 3924–3936, https://doi.org/10.1002/hyp.8201, 2011

Benettin, P., Kirchner, J., Rinaldo, A., and Botter, G.: Modeling chloride transport using travel time distributions at Plynlimon, Wales, Water Resources Research, 51, 3259–3276, https://doi.org/10.1002/2014WR016600, 2015

Wang, S., Hrachowitz, M., Schoups, G., and Stumpp, C.: Stable water isotopes and tritium tracers tell the same tale: no evidence for underestimation of catchment transit times inferred by stable isotopes in StorAge Selection (SAS)-function models, Hydrology and Earth System Sciences, 27, 3083–3114, 2023.

---

## Author Comment (AC2)

**Reviewer 2**

**Comment:** This is a very interesting paper that addresses the elusive role of preferential flows by introducing an additional consideration to the bucket model, which distinguishes the role of preferential flows through shape functions. It eloquently poses the paper's main questions and refers to some of them at the end, thereby focusing the paper on specific aspects of the model and its ability to capture the role of preferential flows. In that respect, this paper has a substantial contribution to the field and should eventually be accepted for publication. However, this is not a "stand-alone paper" as, without reading the previous work by Turk 2024, there is no way to establish the relation between the "bucket ratios" and the model without going over Eqs. 1-26 in the said paper. The submitted paper continues the model from Turk 2024, starting from Eq. 33, and proceeds to examine the sensitivity to the SAS function optimization, specifically to the sensitivity to the preferential flow in light of the parameter (shape function) in the said equations. However, this sensitivity is hard to understand without the full model and figures 3 and 4 in Turk 2024. Further figures are needed from Bogena et al. 2015 (figures 1 and 2) and Turk 2024 (figure 1) to visualize and understand the layout of the field and its relation to the bucket model in Figure S1. I would argue that this paper is a continuation of Turk 2024, model-wise, and it can only be understood if one reads the 2024 paper prior to this paper, and as such, narrows the potential readership of this interesting and relevant paper.

**Response:** We thank Reviewer #2 for the positive feedback on our manuscript. We will address the concerns by providing additional background material in the Supplement. Below, we outline how we consider the issues raised by the reviewer and the changes we intend to make. Reviewer comments are shown in black, while our responses are shown in blue. Specifically, we will (i) briefly summarize the relevant parts of the model setup from Turk (2024), including the bucket ratios and key equations, and (ii) add figures adapted from Bogena et al. (2015) and Turk (2024) to illustrate the field layout and its relation to the bucket model. This will ensure that the present manuscript can be read independently without requiring the reader to consult the previous work.

**Comment:** Furthermore, coming from the transport in soil and rock community, the model seems very phenomenological in nature, which I guess is standard for "bucket models". While it makes a nice attempt to move away from the data-driven model and statistical approaches towards more mechanistic observation, there is no explicit attempt to present the causality between physical parameters to observations. This has to be done when we account for preferential flows, as these preferential flows are the outcome of specific conditions (permeability heterogeneity of the root area and groundwater, head differences within the groundwater stemming from the slope, and slope runoff on a local scale). However, the discussion on these aspects is limited to Section 4, and even there, it does not provide specific details. They are found to some extent in Bogena (2015) for Wüstebach, yet I couldn't find them for the HOAL in Turk (2024). Why is that?

This approach stays on the data side, and while the correlation it provides is an important step towards a more rigorous model, can't there be an effort to hypothesize about the mechanisms that form the varying dominance of preferential flows and see if the data and the missing parameters might corroborate this hypothesis?

**Response:** We acknowledge that our approach is phenomenological and will add a short discussion in Section 4 to link literature-based findings on soil texture, moisture dependence, and slope effects (e.g., Bogena et al., 2015 for Wüstebach, and additional references for HOAL) to the SAS parameterization used here. While our study does not explicitly test mechanistic functions, highlighting these connections will help frame how the SAS shape parameter may reflect underlying physical controls and point to directions for future mechanistic work.

**Comment:** To be more specific in my request, let's take the following example: Is there a possibility to search the literature and provide a functional form that relates how clayey soils under variation in saturation contribute to the emergence of preferential flows, and how this functional form relates to the shape function in this study? Can we compare the sensitivity of this functional form to these aspects (clay concentration, saturation) and the sensitivity of the shape function? Can it account for the model's sensitivity to the shape function? The same can be done for the differences in slope between the two data sets presented in this study. While this is not a mechanistic model, it indicates what parameters are needed

in a mechanistic model, and the sensitivity to them. This will allow this community, and adjacent communities like soil and aquifers, to frame their mathematical approaches in the context of this excellent problem, especially given the excellent datasets presented here. Datasets that urge for a more rigorous and mathematical modeling approach, in space and time, that require some input on the mechanistic origin for these probability density functions controlling the current buckyeet model.

**Response:** Indeed it would be interesting to compare the SAS function shape to different catchment properties, which could allow an a priori parameterization of the SAS function without calibration. In our opinion, this would require testing the model in other catchments that cover a wide range of climates, soils, and topographies, followed by statistical analysis to identify the influential factors controlling SAS function shape. For instance, at the lysimeter scale, Asadollahi et al. (2020) showed that the SAS function shape was similar to the analytical solution of the advection–dispersion equation. However, applying such mechanistic or semi-mechanistic relationships at the catchment scale remains challenging in the context of a "bucket model," as processes are aggregated over the entire system. We will add a short discussion paragraph to highlight this as an open question and future research need, which could provide a bridge between empirical SAS modeling and mechanistic approaches.

Line 31: "In the light of…" or "In light of…"?

**Response:** We will revise the phrase to "In light of …"

**Comment:** The introduction is very good, and so are the questions which represent the study aim and objective.

**Response:** We thank the reviewer for the positive feedback on the introduction and the formulation of the study questions.

**Comment:** Figure 1. Can you provide insight on the streamflow? Maybe maps of the catchment with the locations of the sampling locations? That will help in understanding the significance and time lag of these measurements.

**Response:** We will add maps of both catchments showing the stream network and the locations of the sampling points to Figure 1 (or as a supplementary figure).

**Comment:** Figure S1: TT should be Tr by table S 1. One must define the parameters in black.

**Response:** We will correct Tr to TT in Table S1. In addition, we will clarify that $T_T$ refers to the threshold temperature for snowmelt that needs to be calibrated.

**Comment:** "The darker blue box (Ss,a)" should be Ss,p Not all symbols are defined in Table S 1, please re-check. Maybe move Table S 1 next to Figure S 1 so it will be easier to go between them.

**Response:** We will correct the label to Ss,p and carefully check Table S1 to ensure that all symbols used in the figures and text are consistently defined.We will move the Table S 1 next to the Figure S 1.

**Comment:** Line 167: α and β, do not appear in Table S 1 or Figure S 1, please define them in the right context.

**Response:** We will revise the text and Table S1 to include definitions of α and β. Specifically, we will clarify that β is fixed as 1 for all fluxes, while α is fixed as 1 for preferential groundwater flow but allowed to be calibrated for the unsaturated zone preferential flow (SU,α).

**Comment:** The number of references to Figure S1 suggests that it should be included in the main paper rather than the supplementary material.

**Response:** We will move Figure S1 from the Supplement to the main manuscript.

**Comment:** Equations 1 and 2: How do you define Sr,max? Is it local in time (the maximum of the soil moisture for the given rain event) or is it simply the saturation value, namely the porosity? From the equation, its unclear how this term captures the preferential flow, specifically how it relates it to physical aspects, which are not saturation (heterogeneity and head differences)

**Response:** We will clarify that Sr,max is a calibrated parameter representing the maximum root-zone storage capacity, not porosity. The SAS function for age selection of preferential flow from the unsaturated root zone (Rf, Fig. S1) was formulated as a time-variable function of relative soil wetness to reflect changes in transport processes between wet and dry soil conditions. In particular, it accounts for the preferential release of younger water as soil wetness increases, consistent with previous studies showing that wetter soils promote faster flow paths with reduced mixing of water. The temporal variability in the SAS function was implemented through the time-dependent shape parameter $\alpha(t)$ (Eq. 2).

**Comment:** Line 223-226: Since there is no discussion on how preferential flow components like heterogeneity and head differences were implemented in the model, it is hard to understand if this is the right way to measure the model's sensitivity to it.

**Response:** We will clarify the model setup and revise the description of the second test to emphasize that our approach is based on age-ranked groundwater storage. Specifically, the section will be revised to:

"In the model setup, groundwater storage was formulated as consisting of an 'active' groundwater storage (Ss,a) and a hydrologically 'passive' storage volume (Ss,p, mm). The passive storage does not change over time if there are no deep infiltration losses (Zuber, 1986; Hrachowitz et al., 2016; Wang et al., 2023). While Ss,p does not contribute directly to runoff, it isotopically mixes with water in the active storage, thereby influencing the tracer signal of the outflow. The total groundwater storage (Ss,tot) was therefore considered as the sum of Ss,a and Ss,p, so that the age-ranked groundwater storage (Ss,tot) inherently reflected a mixture of these storage volumes. Next, we tested the sensitivity of the $\delta^2H$ signal to changes in the groundwater SAS function (Fig. 2b) by varying $\alpha$ across the same range—0.1, 0.7, 1.0, and 5.0— while fixing the previously optimized $\alpha_0$ value for the root zone. This second test was designed as an age-ranked experiment to examine if (and how) preferential release of younger groundwater influences transit time distributions and tracer simulations. For clarification, the SAS formulation can only identify whether preferential release of young water occurs, but it does not resolve the underlying physical mechanisms that generate it."

**Comment:** Figure 2: Why is the CDF shown and not the PDF for the TTD? Indeed, one is the derivative of the other, yet the PDF allows us to see the tailing and the mean behavior more clearly. I'm also not sure if eCDF is the cumulative of , as it is not stated, and from the curvature, it's hard to establish it.

**Response:** We will clarify in the text and figure caption that eCDF refers to the empirical cumulative distribution function. We selected cumulative distributions to show the individual curves more clearly, and for scenario comparison we found cumulative curves easier to interpret than PDFs.

**Comment:** A non-intuitive aspect is that preferential release of older water ($\alpha>1$) has the highest residence time. I assume that this is due to the interplay between the ground and root area flow, which affects the fast response water in Figure S1. Therefore, as $\alpha$ increases above 1, less water is directed to the fast response bucket; however, there is no indication of this. Where are the equations related to each "bucket"? How do you parameterize these "buckets"?

Another aspect is that Figures 2a, b, and 3 are basically identical. I understand that they are "illustrations", but if there is no difference among the illustrations, what is the point of showing 3 of them? Furthermore, the illustration of the different passive storage values does not provide any insight into how the analysis differs or how the results should change as a result.

**Response:** We thank the reviewer for this comment. We will improve the clarity of both the model description and the figures.

To address the first point, we will clarify in the methods section that preferential release of older water ($\alpha > 1$) increases mean residence times because less young water is directed toward the fast-response

pathways (root-zone preferential flow), leading to a stronger weighting of slower groundwater contributions. We will also add the relevant bucket-model equations for water age balance equations and (currently included in Turk 2024) section "Integration of the rSAS function concept and the hydrological model", and provide a clearer description of how each bucket tracer balance through SAS formulation is parameterized.

To address the second point, we will streamline the figures to avoid redundancy. Specifically, we will keep only the necessary illustrations and clarify in the captions what information each figure is intended to convey. For passive storage values, we will highlight in the results section how varying $S_{s,p}$ affects tracer damping and the tails of the TTD, which was the purpose of including those scenarios.

**Comment:** Line 248: 15 parameters are identified in Table S1, and I guess there are multiple locations at which they are measured, which spans the calibration to 55 and 190 parameters, but instead of guessing, please clarify this aspect in a text, or better yet, a figure outlining the measurements in the catchment area.

**Response:** We will clarify in the text that the 15 parameters listed in Table S1 were calibrated for each catchment. The number of parameter solutions (55 for HOAL and 190 for Wüstebach) refers to the sets of parameter combinations that satisfied the calibration criteria (DE < 1), not to the number of calibrated parameters. To avoid confusion, we will add a schematic figure of each catchment that outlines the main measurement locations (precipitation, streamflow, isotopes) and indicate how these data were used in the calibration.

**Comment:** Line 278: "In the HOAL catchment, the calibrated root-zone SAS shape parameter lower bound $\alpha_0 = 0.14$ indicated a strong preference for very young water through unsaturated root zone preferential flow pathways, suggesting that precipitation rapidly reached the stream with minimal mixing with stored water." While the correlation is clear, I'm not sure I understand the physical aspect of it. The unsaturated root zone is controlled by capillary forces if it is indeed unsaturated and stagnant, and by the connected paths of saturated areas (or preferential flows) under unsaturated flow. As such, the mechanism I can envision is that when starting with dry conditions and high infiltration, the invaded paths, which form the latter as preferential flows, must be occupied by younger water. However, under higher saturation, preferential flows are already established within the root zone; therefore, at lower infiltration rates, the discharge will consist of older water. Is this the physical mechanism suggested by this finding? If so, please refer to it; otherwise, what is the conclusion I need to draw from this finding, and how is it related to questions 1 and 2? This aspect is discussed in lines 364-372, where the difference between the clay soil of HOAL, which promote young water through the preferential flows, in contrast to the forest cover soil in Wüstebach which promote a more uniform infiltration, and therefore reduce the preferential flow effect and increases both the residence and the storage in the subsurface which is addressed later in the paper.

**Response:** We thank the reviewer for this comment. We agree that the physical interpretation of the calibrated $\alpha_0$ value requires clarification. We will revise the text to make the connection to preferential flow mechanisms more explicit. Specifically, in the HOAL catchment the calibrated root-zone SAS shape parameter lower bound ($\alpha_0 = 0.14$) indicates that under dry antecedent conditions with intense events, preferential flow paths are quickly activated, allowing younger water to reach the stream with limited mixing. In contrast, under wetter antecedent conditions, established preferential flow pathways facilitate more mixing compared to overland flow, leading to relatively older (older than recent precipitation) water contributions. This interpretation is consistent with the time-variable formulation of $\alpha$ (t) (Eq. 1), where the parameterization links soil wetness and precipitation intensity to the activation of preferential flows. We will also clarify that this finding directly relates to research question 1 (whether tracer data can identify preferential flow) and question 2 (how preferential flows influence transit time distributions).

**Comment:** Figure 6. Was there an attempt to relate the PDF (the derivative of the CDF) to the preferential flow specific length and time scales using the specific field data?

**Response:** No, we did not attempt to relate the PDF to preferential flow length and time scales using field data in this study, as this would require additional field measurements that go beyond the scope of the present work.

**Comment:** Line 316-320: In the spirit of my previous comment, the mechanism here is that it is solely controlled by the "pipeline of the preferential flows" within the groundwater, and the storage size represents the pipeline length and the extent or dominance of groundwater preferential flows? If this is the case, do we have indications for this for the HOAL and Wüstebach? Do we see that the first is more heterogeneous and therefore is more likely to lean towards preferential flows than the latter? Alternatively, it may stem from the difference in slope, where the first is steeper than the latter (as is clear from Figure 1 in Turk 2024 and Bogena 2015, respectively), which will exacerbate the inherent heterogeneity, leading to more preferential flows and shorter storage times for the HOAL case.

This is partially addressed in lines 345-350; however, the mechanism by which the dominance of preferential flows is related to the different conditions of each field is provided later in lines 364-372, favoring the first explanation. However, the second explanation is partially supported by the passive groundwater storage difference between these fields (lines 392-396). Are they both true? Equal?

**Response:** Our model formulation does not explicitly represent the physical "pipeline" of preferential groundwater flow, nor does it resolve heterogeneity or slope effects mechanistically. Rather, the sensitivity analysis with different passive storage volumes provides a phenomenological way of testing how much tracer damping could result from larger or smaller passive storage contributions.

We will clarify in the discussion that both mechanisms likely contribute: (i) soil/heterogeneity and slope control the activation and connectivity of preferential pathways, and (ii) passive groundwater storage controls the damping and persistence of tracer signals (and thus the residence times). Our data support the presence of both effects but do not allow us to quantify their relative contributions. We will revise the text to state this explicitly and to frame the two as complementary hypotheses rather than competing explanations, noting that disentangling them would require additional filed observations.

**Comment:** These comments on the mechanism of the preferential flows and their sources are reflected in the following (lines 412-416): "An alternative explanation, however, must also be considered: it is possible that such preferential groundwater flow processes are simply absent or negligible in the HOAL and Wüstebach catchments. The current data and model structure are insufficient to conclusively rule out either possibility. Ultimately, distinguishing between limitations in model sensitivity and the actual absence of preferential flow processes requires additional, spatially distributed tracer data and complementary hydrometric observations." As there are no hypotheses on the mechanism leading to preferential flows that stem from the physical aspects of the fields —permeability heterogeneity of the root area and groundwater, head differences within the groundwater stemming from the slope, and slope runoff on a local scale —one cannot draw a clear conclusion. These physical considerations are evident only at the correlational level from the data, as is clear from the Turk 2024 model, and not from a causality standpoint, which would require a more rigorous consideration. This is a major weakness of this study, and an attempt to formulate a hypothesis that will drive others to see what the actual causality is is missing. Nonetheless, this statistical approach is an important step toward a rigorous model, as it points to the conditions under which preferential flows can be more dominant. An example of how the statistical approach may point towards a more rigorous model is found in lines 430-438, where the distribution tail points to the ratio between active and passive storage.

**Response:** We thank the reviewer for this comment. We agree that our current framework cannot resolve the causal mechanisms of preferential flow processes, and that this represents a limitation of simpler, top-down catchment-scale, isotope-based transport models. We will expand the discussion (lines 412–416) to clarify this point and to formulate ideas that may guide future work. Specifically, we will note that (i) soil heterogeneity and slope likely govern the activation and connectivity of preferential flow pathways, (ii) differences in passive versus active storage modulate tracer damping and the apparent age distributions, and (iii) the relative importance of these processes remains unresolved with the current data.

"Our analysis does not allow us to establish causal mechanisms of preferential flow. However, based on previous field evidence, factors such as soil heterogeneity, slope, and passive storage volumes may play a role in modulating the strength of preferential flow signals. While these mechanisms remain hypotheses beyond the scope of top-down catchment-scale, isotope-based transport models, the SAS-based framework highlights conditions under which such effects may become detectable."

**Comment:** Discussion: I extremely like how the authors pose the three questions at the beginning of the paper, yet they only return to the first two at the end of the paper. Even if the conclusion is that it is inconclusive, this should appear for completeness.

**Response:** We thank the reviewer for this suggestion. We will revise the Discussion and Conclusions to explicitly return to all three research questions. For the third question, we will emphasize that passive storage volume and associated mixing assumptions introduced substantial variability in the estimation of transit time distributions, particularly in the long tails. However, the current data and model structure do not allow us to resolve this variability conclusively, and we will highlight this as a key limitation in catchment scale lumped transport model.

Line 399: add a space after groundwater.

**Response:** We will add a space after groundwater.